# A Study of Quantum Game for Low-Carbon Transportation with Government Subsidies and Penalties

**Yongfei Li** [1], **Jiangtao Wang** [1], **Bin Wang** [2,*] and **Clark Luo** [3]

[1] School of Modern Post, Xi'an University of Posts and Telecommunications, Xi'an 710061, China; lyfking2000@163.com (Y.L.); 15591095873@163.com (J.W.)
[2] College of Business and Public Management, Kean University-Wenzhou Campus, Wenzhou 325060, China
[3] Waikato Institute of Technology, Hamilton 3204, New Zealand; clark.luo@wintec.ac.nz
[*] Correspondence: wbill@kean.edu

**Abstract:** Traditional classical game theory struggles to effectively address the inefficiencies in subsidizing and penalizing the R&D and production of low-carbon transportation vehicles. To avoid the shortcomings of classic game theory, this research integrates quantum game theory with Nash games to explore the characteristics of automakers' behavior for low-carbon transportation with government subsidies and penalties. We first constructed a low-carbon transportation game model between the government and automakers. Then, the optimal profit strategies for both parties in a quantum entangled state were analyzed. Finally, the impact of quantum superposition states and the initial entangled state on the profit strategies of both parties was simulated and analyzed using Monte Carlo simulations. We find that under the joint effects of government subsidies and penalties, quantum game states and the initial quantum entangled state have a crucial influence on the game's outcomes. They can encourage the realization of Nash equilibrium and perfect coordination in the quantum game, significantly increasing the profits for both parties. This in turn effectively stimulates automakers to research and produce low-carbon transportation solutions, promoting the rapid development of low-carbon transportation technology. In theory, this research can enrich the Quantum game for improvements in the Nash equilibrium solution for the government to subsidize and penalize the low-carbon transportation problem. Meanwhile, in practice, it can provide guidance and reference in optimal strategy selection conditions for government policymakers and automakers for low-carbon transportation.

**Keywords:** low-carbon transportation; automakers; quantum game theory; Nash equilibrium theory

## 1. Introduction

The transportation sector, as one of the major sources of global $CO_2$ emissions, accounts for approximately 24% of the total $CO_2$ emissions, with most of the emissions coming from fossil fuel vehicles [1]. To address the challenges of carbon emissions, carbon neutrality, and peak carbon, governments and numerous automakers worldwide are highly concerned about the development of low-carbon transportation. The global society is gradually stepping into a new era of low-carbon transportation, with various countries and regions implementing policies to promote the transition to low-carbon vehicles [2,3]. For instance, the European Union's Green Deal stipulates a minimum reduction of 55% in carbon dioxide emissions from the transportation sector by 2030. The average emissions of new vehicles are required to decrease by 55%, starting from 2030, and by 100% from 2035 onwards (https://zh-cn.eureporter.co/environment/european-green-deal/ (accessed on 10 January 2024)). The Netherlands plans to achieve zero emissions for all new vehicles by 2030 (http://nl.mofcom.gov.cn/article/ztdy/202301/20230103379939.shtml/ (accessed on 10 January 2024)), France intends to ban the sale of gasoline-powered cars by 2040 (https://unfccc.int/news/france-aims-to-end-sale-of-fossil-fuel-powered-cars-by-2040/ (accessed on 10 January 2024)), Sweden's target is to achieve net-zero emissions by

no later than 2045 (https://www.government.se/articles/2021/05/sweden-mobilises-to-electrify-regional-freight-transport/ (accessed on 10 January 2024)), and Germany plans for at least 15 million newly registered cars to be electric by 2030, representing half of the total (https://www.euractiv.com/section/electric-cars/news/new-german-government/ (accessed on 10 January 2024)). China has also implemented multiple government subsidies and penalties for automakers, committing to peak carbon emissions by 2030 and carbon neutrality by 2060 (http://english.scio.gov.cn/whitepapers/2021-10/27/ (accessed on 10 January 2024)). Nordic countries have implemented policies such as tax reduction and free charging to promote the purchase of electric vehicles and have also imposed some tax measures to increase the cost of using fossil fuel vehicles.

Despite the various government subsidies and penalties imposed on automakers to promote the development of low-carbon transportation, companies such as Honda, Volkswagen, Toyota, Ford, and Changan continue to develop and produce both fossil fuel and low-carbon vehicles due to their practical interests, engaging in multiple negotiations with governments. There exist practical issues, such as inefficiencies in government subsidies and penalties, for automakers in the R&D and production of low-carbon transportation vehicles, as well as a lack of enthusiasm among automakers for the R&D and production of low-carbon transportation vehicles. Consequently, the effectiveness of government subsidies and penalties is not always evident [4–9]. This research aims to address the aforementioned real-world issues. It is imperative to conduct thorough investigations into the game theory of low-carbon transportation under the combined influences of government subsidies and penalties. This comprehensive approach is essential for expediting the transformation and advancement of these enterprises toward the realm of low-carbon transportation development.

However, current research on government subsidies and penalties related to low-carbon transportation primarily relies on traditional economic game theory methods [10–17]. Classical game models often fail to attain pure Nash equilibrium solutions and overlook the potential interdependencies among the players in their strategic choices. They also neglect the impact of preference selection for more favorable strategies on their payoffs. The shortcomings of classical game theory occasionally result in a lack of equilibrium solutions and optimal solutions. As a result, the effectiveness of the game is significantly constrained, making it challenging to effectively incentivize automakers to research and produce low-carbon transportation vehicles [14,15]. Therefore, seeking a method more suitable for addressing the game-theoretical challenges in low-carbon transportation is a key objective of this study. This study primarily focuses on the theoretical aspects of quantum game theory, aiming to address issues through deductive reasoning and analysis. The aim of this research is to explore the potential application of quantum game theory methods in the field of transportation, distinguishing itself from traditional classical game theory methods. This endeavor seeks to offer novel perspectives and methodologies for studying and addressing transportation issues.

Some scholars have found that quantum games introduce a new element of randomness, leading to more diverse game strategy choices and more complex game dynamics compared to classical games. Quantum entanglement, a unique feature of quantum systems, allows for the establishment of a special relationship between two or more quantum systems, making them function as a whole to create new strategies and possibilities. This phenomenon facilitates Nash equilibrium and optimal choices, addressing the problem of the absence of Nash equilibrium solutions in traditional classical games [18,19]. For example, if two electrons have opposite magnetic moments, they are in an entangled state. This means that if we measure the magnetic moment of one electron and find it to be upward, the magnetic moment of the other electron must be downward. In this example, the two electrons can be seen as a whole, and they have a special mutual connection. When we measure one electron, the state of the other electron is also influenced. This is the essence of quantum entanglement. At present, research on quantum games has been conducted in various fields, including economics and market analysis, resource allocation

and cooperation, network and social media analysis, policy formulation and international relations, environmental sustainability, finance, and risk management. Hence, quantum game theory is considered a more suitable approach for addressing the game-theoretical challenges in low-carbon transportation.

Our research considers the significant joint effects of quantum game theory and Nash equilibrium theory on the issue of low-carbon transportation. We have developed a novel game model considering the combined effects of government subsidies and penalties on automakers. This model particularly focuses on the influence of quantum superposition states and quantum initial entangled states on the low-carbon strategies between the government and automakers. Quantum superposition represents the uncertainty of the government in its policies regarding low-carbon subsidies and penalties, as well as the development and production of transportation vehicles by automakers. On the other hand, quantum entanglement reflects the shared interests or competitive dynamics between the government and automakers in environmental protection. It clarifies the significant impact of the synergy between government subsidies, penalties, quantum initial entangled states, and quantum superposition states on the profits of both the government and automakers. This model creates new game strategies and facilitates more stable Nash equilibrium and optimal choices. This research solves the problem of the absence of Nash equilibrium solutions in traditional classical games. This research contributes to enhancing the profits of both the government and automakers, promoting the development of low-carbon transportation, alleviating environmental pressures, and making contributions to the research and practice of global low-carbon transportation.

This research is divided into four parts: model description, model construction and analysis, simulation analysis, and conclusion and insights. In game theory models, it is common to simplify the model to make it easier to analyze and understand. Drawing upon the literature review of numerous scholars, this paper focuses on the following parameters in the model: subsidy amount, penalty amount, sales revenue, R&D costs, tax revenue, and environmental losses.

## 2. Literature Review

### 2.1. Government Subsidies and Penalties for Automakers in Low-Carbon Research

The transportation sector accounts for 24% of global carbon dioxide emissions, with vehicle exhaust emissions being a major contributor [1]. Consequently, governments have formulated a series of policy objectives aimed at reducing transportation emissions. These objectives include lowering vehicle exhaust emissions and promoting low-carbon travel modes [7]. However, achieving these objectives is not straightforward. Technologically, further measures such as the promotion of new energy vehicles are necessary, while economically, considerations of investment costs, benefits, and government fund allocation are crucial [20]. Therefore, governments need to comprehensively consider various factors to devise practical and feasible policy measures to achieve emission reduction targets. Researchers have extensively studied the effectiveness of these government policies and have put forth several key insights. This paper focuses on reviewing government subsidy and penalty policies.

Shao et al. (2021) conducted a study on 88 Chinese new energy vehicle companies, concluding that research and development subsidies have a significant incentivizing effect on R&D activities [4]. However, the diminishing marginal effect of subsidy intensity on the companies' R&D activities suggests a shift from subsidies to penalties. Li et al. (2020) analyzed the reduction of subsidy policies' significant impact on the adoption rate of electric vehicles, emphasizing the need to consider policymakers' preference choices in policy formulation [5]. Yang et al. (2021) evaluated the shortcomings of government policies regarding subsidies and penalties on new energy vehicle companies using text analysis [6]. Fritz et al. (2019) conducted empirical research on the strict fuel penalty policy's impact on carbon emissions from electric vehicles and highlighted the inadequacies in government formulation [7]. Hasan et al. (2020) examined the acceptability of carbon emission reduction

through subsidies and carbon tax penalties using multiple analytical methods, finding that most policies were unpopular [8]. Ou et al. (2019) compared the status of the light plug-in electric vehicle market in China under government subsidies to that in the United States, revealing that top traditional Chinese automakers were relatively less proactive [9]. The studies mentioned above suggest that the effectiveness of government subsidies and penalties is not entirely clear. Subsequently, many scholars have explored the interaction between the government and automakers and how they affect policy outcomes, using classical game theory methods.

*2.2. Low-Carbon Transportation Game Theory Considering Government Subsidies and Penalties*

Scholars have primarily focused on government subsidies and penalties in the low-carbon transportation game research in three main areas: Stackelberg non-cooperative games, evolutionary games, and complex network games.

(1) In the context of Stackelberg non-cooperative games, Zhao et al. (2020) investigated the changes in automakers' profits under different government subsidies but did not consider the government's punitive measures against automakers [10]. Fan et al. (2020) studied the impact of government subsidies and tariff penalties on the electric vehicle market, focusing on how the technological development of electric vehicles influences the government's optimal subsidy decisions but did not account for the punitive effects of government subsidies default by automakers [11]. Srivastava et al. (2022), aiming to increase the market penetration of electric vehicles, compared and analyzed different game models under government subsidies and no subsidies, as well as uniform and differential carbon tax penalty policies for electric vehicle manufacturers. However, their tax penalties were aimed at levying green taxes on electric vehicle manufacturers and did not consider penalties for fuel vehicle manufacturers for not developing low-carbon transportation solutions [12].

(2) In the context of evolutionary games, Wang and Li (2023) analyzed the impact of government regulations on the development of hydrogen fuel cell vehicles. They argued that the more severe the penalties imposed by the government on pollution caused by hydrogen fuel cell vehicle manufacturers, the more likely these manufacturers are to produce hydrogen fuel cell vehicles. The larger the research and development subsidies, the more favorable it is for the manufacturers. However, their study overlooked the government's revenue and failed to achieve perfect coordination between the government and automakers [13]. Ji et al. (2019) considered establishing subsidy-related mechanisms for local governments in new energy vehicle enterprises. They found that ideally, government subsidies were positively correlated with punishing fuel vehicle manufacturers if they did not develop new energy vehicles. However, in reality, the ideal subsidy effect as proposed by Ji et al. (2019) could be achieved without meeting ideal conditions [14]. Zheng et al. (2023) studied the impact of static and dynamic carbon tax penalties on new energy vehicle manufacturers based on prospect theory. They found that there were no stable points and equilibrium strategies under static carbon tax penalties. Our research shows that quantum games can effectively obtain stable equilibrium solutions [15].

(3) In the realm of complex network games, there are relatively few studies on the government's subsidies and penalties for low-carbon transportation games through complex network games. Among them, Wang et al. (2023) studied how the government could effectively subsidize new energy vehicle manufacturers to maximize the effectiveness of government subsidies, but they did not consider the government's punitive measures against automakers [16]. Zhao et al. (2021) based on probability space, established a complex network game mechanism of local government subsidies and penalties for new energy vehicle manufacturers in four classical networks including scale-free and scale networks, simulating the promotion process of new energy vehicles [17]. However, they did not consider the correlation effects of quantum state superposition and quantum entanglement within the network, which could play a crucial role in the game results.

In summary, the aforementioned traditional classical games are based on probabilistic space game models, which are difficult to accurately portray human decision-making processes involving quantum state superposition and quantum entanglement [21]. Any game in the real world involves information exchange and interaction among game individuals. The decision-making process may not be independent but interdependent. Decision-making processes can be viewed as quantum game information processing processes, which are often more complex than traditional classical games, making it difficult to effectively solve many real-world problems, such as government subsidies and penalties for low-carbon transportation games; we can employ the theory and methodologies of quantum game theory [22–24].

*2.3. Quantum Game*

Theory Research Quantum game theory was first proposed by scholar Meyer et al. (1999), and they, along with other researchers, found that the payoff outcomes in quantum games are generally superior to those in classical games, or at least not worse [18,19,25,26]. Du et al. (2002) successfully conducted a "Quantum Prisoner's Dilemma" experiment using nuclear magnetic resonance [27]. In 2016, physicists launched the world's first quantum science experiment satellite, "Micius", providing an opportunity for the development of quantum technology [28]. Haven and Khrennikov (2013) co-authored "Quantum Social Science", attempting to explain how quantum game theory can be applied to research in psychology, decision-making, finance, economics, and other social sciences [29]. Albert (2015) published the book "Quantum Mind and Social Science: Unifying Physical and Social Ontology", with an attempt to expound upon issues related to human consciousness and social phenomena using quantum game theory [30]. Several scholars have also contributed a series of articles employing quantum game theory in the field of social science. Zhang et al. (2020) conducted explorations in the realm of marketing in economics [31]. Shi et al. (2021) analyzed market demand functions in economics [32]. Samadi et al. (2018) delved into resolving the issue of temporal inconsistency in economics [33]. Liu et al. (2023) provided recommendations on how to promote the synergy of investment and lending [34]. Herman et al. (2023) discussed the advantages and limitations of the financial sector [35]. Schneckenberg et al. (2023) explored how businesses can reconcile the paradox of making both virtuous and controversial decisions in their strategic choices [36].

There is relatively limited research on quantum games in the field of low-carbon transportation. Some scholars have used quantum game methods to study the design of green logistics networks in multimodal transportation [37]. However, there is no evidence of the use of quantum game theory to address the low-carbon transportation game involving government subsidies and penalties for automakers. Currently, there is limited literature on quantum game theory in the entire field of economic management activities, primarily due to two main reasons. On one hand, quantum game methods are relatively complex and require specific background knowledge, limiting their use by some researchers. On the other hand, quantum game theory is fundamentally based on quantum mechanics, which is still evolving, making it challenging to observe quantum effects in practical economic and management decision-making. Nevertheless, quantum games offer certain advantages compared to traditional classical games. Based on these considerations, using quantum game theory to study the low-carbon transportation game involving government subsidies and penalties for automakers is a viable approach.

## 3. Model Description

This study considers the quantum game of low-carbon transportation under the combined effects of government subsidizing and penalizing for automakers. Let us denote the government as A and the automakers as B, both of which do not require full rationality and independent decision-making. The government's strategy choices for promoting the R&D and production of low-carbon transportation solutions are labeled as *V* (subsidize and penalize) and *O* (not subsidize and not penalize), while the automak-

ers' strategy choices for producing low-carbon transportation solutions are denoted as
$N$ (R&D and production) and $M$ (not to R&D and production). The quantum strategy
set for the government is represented as $\{|V\rangle, |O\rangle\}$, and the quantum strategy set for the
automakers is represented as $\{|N\rangle, |M\rangle\}$. The four-dimensional Hilbert space is given by
$H = \{|VN\rangle, |VM\rangle, |ON\rangle, |OM\rangle\}$. Assuming that the government chooses the strategy $V$
with a probability of $x$, then the probability of the government choosing the strategy $O$ is
$1 - x$. Similarly, if the automakers choose the strategy $N$ with a probability of $y$, then the
probability of the automakers choosing the strategy $M$ is $1 - y$. Here, $x^*$ and $y^*$ represent
the mixed Nash equilibrium probabilities for the government and automakers, respectively.

The government imposes a one-time carbon tax and fuel tax on consumers when they
purchase non-low-carbon transportation vehicles (primarily fuel-powered cars), with a
tax amount $a_1$ per vehicle [38]. Automakers undertake low-carbon transportation vehicle
R&D and production measures; they earn revenue $a_2$ and incur costs $a_3$ [39]. When au-
tomakers do not R&D low-carbon transportation vehicles, the government incurs a certain
environmental cost $a_4$ per vehicle [38]. Without the R&D and production of transportation
vehicles, automakers earn a profit $a_5$ per vehicle by producing and selling traditional
transportation vehicles [39].To incentivize automakers to undertake R&D and to produce
low-carbon transportation vehicles, the government provides subsidies to them, with each
subsidy amounting to $a_6$ per vehicle [20]. After receiving subsidies from the government, if
automakers do not undertake R&D and produce low-carbon transportation vehicles, they
face a fine $k$ per vehicle [40], where $k > a_6$. The payoff matrix under traditional classical
game conditions can be obtained, as shown in Table 1.

**Table 1.** Payoff matrix under traditional classical game conditions.

| Payoffs | | B | |
|---|---|---|---|
| | | **N** | **M** |
| A | $V$ | $(a_1 - a_6, a_6 + a_2 - a_3)$ | $(a_1 + k - a_4 - a_6, a_5 + a_6 - k)$ |
| | $O$ | $(a_1, a_2 - a_3)$ | $(a_1 - a_4, a_5)$ |

Government A and automakers B's quantum game takes place in a Hilbert space, with
quantum states denoted as both the left ket $\langle\varphi| = (a^*, b^*)$ and the right ket $|\varphi\rangle = \begin{pmatrix} a \\ b \end{pmatrix}$.
Here, $a$ and $b$ represent the entanglement under different strategies, and $a^*$ and $b^*$ are
their complex conjugates. In the context of quantum game theory, the basic unit is the
quantum bit, where a single quantum bit has two states, represented as $|0\rangle$ and $|1\rangle$. The
Dirac notation $|\rangle$ is used for quantum states, with a quantum superposition state defined as
$|\psi\rangle = a|0\rangle + b|1\rangle$, satisfying the condition $|a|^2 + |b|^2 = 1$. Here, $|a|^2$ and $|b|^2$ represent the
initial entanglement state under a single quantum bit [41].

Based on the need for a quantum game between both government A and automakers
B, to extend the concept of single quantum bits to double quantum bits, use $a$, $b$, $c$, and $d$ as
measures of the quantum entanglement for strategies $|VN\rangle$, $|VM\rangle$, $|ON\rangle$, and $|OM\rangle$ under
a double quantum bit scenario, satisfying the condition $|a|^2 + |b|^2 + |c|^2 + |d|^2 = 1$ [33]. $a$
and $b$ are used to assess the choices made by government A for strategies $V$, and automakers
B's choices for $N$ and $M$. $c$ and $d$ are used to evaluate the decisions of government A
for strategies $O$ and automakers B's choices for $N$ and $M$. $|VN\rangle$ strategy represents the
government adopting subsidy and penalty measures, and automakers R&D and produce
low-carbon transportation vehicles. $|VM\rangle$ strategy represents the government adopting
subsidy and penalty measures, and automakers do not R&D and produce low-carbon
transportation vehicles. $|ON\rangle$ strategy represents the government not adopting subsidy
and penalty measures, and automakers R&D and produce low-carbon transportation
vehicles. $|OM\rangle$ strategy represents the government not adopting subsidy and penalty
measures, and automakers do not R&D and produce low-carbon transportation vehicles.

From Table 1, it can be observed that $|a|^2 + |b|^2$ corresponds to the entanglement under the government's subsidy and penalty strategy, $|c|^2 + |d|^2$ corresponds to the entanglement under the government's no-subsidy and no-penalty strategy, $|a|^2 + |c|^2$ corresponds to the entanglement under the automakers' R&D and produce strategy, and $|b|^2 + |d|^2$ corresponds to the entanglement under the automakers' non- R&D and produce strategy. It is important to note that the initial entanglement states $|a|^2$, $|b|^2$, $|c|^2$, and $|d|^2$ will undergo a collapse phenomenon after being observed by decision-makers from both the government and automakers. The initial entanglement states will transition into superposition states and will exhibit mutual entanglement, affecting the decisions and payoffs of both parties. The tensor product in quantum game theory is denoted by $\otimes$, and the Hermitian conjugate of an operator $A$ is denoted as $A^\dagger$, which can alter the states of the left ket and the right ket [42]. Without loss of generality, the quantum model employs the widely used Marinatto–Weber model as shown in Figure 1 [41].

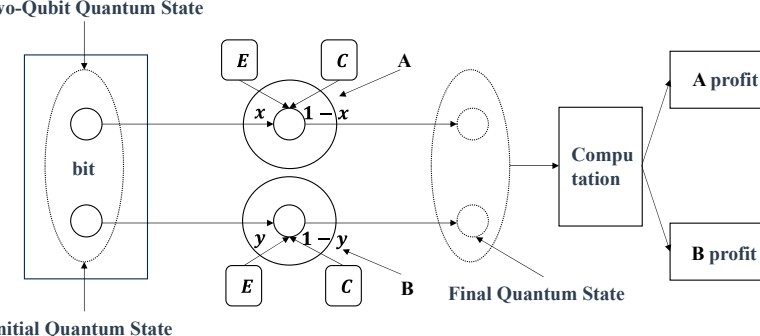

**Figure 1.** Schematic diagram of the Marinatto–Weber model.

In the quantum Marinatto–Weber model, the computational procedure is referred to as the quantumization process. The quantumization process involves three steps: In the first step, an outer product operation is performed using the initial state $|\varphi_{in}\rangle$ and its dual state $\langle \varphi_{in}|$ to obtain the initial density matrix $\rho_{in}$. In the second step, both parties choose quantum bit identity operator $E$ and quantum bit flip operator $C$ with probabilities $x$, $1-x$, $y$, and $1-y$. Here, $E = \begin{pmatrix} 1 & 0 \\ 0 & 1 \end{pmatrix}$, $C = \begin{pmatrix} 0 & 1 \\ 1 & 0 \end{pmatrix}$. Applying local transformations to the initial density matrix $\rho_{in}$ using these operators results in the final density matrix $\rho_{fin}$. In the third step, the trace of the product of the final density matrix $\rho_{fin}$ with the profit operators $P_A$ and $P_B$, respectively, yields the respective profits for both parties, $\pi'_A(x,y)$ and $\pi'_B(x,y)$, i.e., $Tr(P_A \rho_{fin})$ and $Tr(P_B \rho_{fin})$.

The economic implications brought about by this model can be understood by referring to the following literature, aiding readers in comprehension [25–36].

## 4. Construction and Analysis of the Quantum Game Model

*4.1. The Construction of the Quantum Game Model*

From the model description, we can derive the initial quantum state $|\varphi_{in}\rangle$ as follows:

$$|\varphi_{in}\rangle = a|VN\rangle + b|VM\rangle + c|ON\rangle + d|OM\rangle \tag{1}$$

From this $|\varphi_{in}\rangle$, we can derive the initial density matrix $\rho_{in}$ as follows:

$$\rho_{in} = |\varphi_{in}\rangle\langle\varphi_{in}| \tag{2}$$

To obtain the final density matrix $\rho_{fin}$, the initial density matrix $\rho_{in}$ should be transformed by applying the identity operator. Let $E$ represent the quantum bit identity operator and $C$ represent the quantum bit flip operator, where $E = \begin{pmatrix} 1 & 0 \\ 0 & 1 \end{pmatrix}$, $C = \begin{pmatrix} 0 & 1 \\ 1 & 0 \end{pmatrix}$. Quantumization is performed separately on the strategies of the government and the automakers,

with the processes being $xE + (1 - x)C$ and $yE + (1 - y)C$, respectively. This results in the final density matrix $\rho_{fin}$ as follows:

$$\rho_{fin} = xy[E_A \otimes E_B]\rho_{in}(E_A{}^\dagger \otimes E_B{}^\dagger) + x(1-y)[E_A \otimes C_B]\rho_{in}(E_A{}^\dagger \otimes C_B{}^\dagger)$$
$$+ (1-x)y[C_A \otimes E_B]\rho_{in}(C_A{}^\dagger \otimes E_B{}^\dagger) + (1-x)(1-y)[C_A \otimes C_B]\rho_{in}(C_A{}^\dagger \otimes C_B{}^\dagger) \tag{3}$$

The quantumized payoff for the government, denoted $\pi'_A(x, y)$, is as follows:

$$\pi'_A(x,y) = Tr(P_A\rho_{fin}) = (a_1 - a_4)((x-1)(y-1)|a|^2 - y(x-1)|b|^2 - x(y-1)|c|^2$$
$$+ xy|d|^2) + (a_1 - a_6)(xy|a|^2 - x(y-1)|b|^2 - y(x-1)|c|^2 + (x-1)(y-1)|d|^2)$$
$$+ (-x(y-1)|a|^2 + xy|b|^2 + (x-1)(y-1)|c|^2 - y(x-1)|d|^2)(a_1 - a_4 - a_6 + k)$$
$$+ a_1(-y(x-1)|a|^2 + (x-1)(y-1)|b|^2 + xy|c|^2 - x(y-1)|d|^2) \tag{4}$$

The quantumized payoff for the automakers, denoted $\pi'_B(x, y)$, is as follows:

$$\pi'_B(x,y) = Tr(P_B\rho_{fin}) = (a_2 - a_3)(-y(x-1)|a|^2 + (x-1)(y-1)|b|^2 + xy|c|^2 - x(y-1)|d|^2) + (a_5 + a_6 - k)(-x(y-1)|a|^2 + xy|b|^2 + (x-1)(y-1)|c|^2 - y(x-1)|d|^2) + (a_2 - a_3 + a_6)(xy|a|^2 - x(y-1)|b|^2 - y(x-1)|c|^2 + (x-1)(y-1)|d|^2) + a_5((x-1)(y-1)|a|^2 - y(x-1)|b|^2 - x(y-1)|c|^2 + xy|d|^2) \tag{5}$$

where, $P_A = U_A(V,N)|VN\rangle\langle VN| + U_A(V,M)|VM\rangle\langle VM| + U_A(O,N)|ON\rangle\langle ON| + U_A(O,M)|OM\rangle\langle OM|$, $P_B = U_B(V,N)|VN\rangle\langle VN| + U_B(V,M)|VM\rangle\langle VM| + U_B(O,N)|ON\rangle\langle ON| + U_B(O,M)|OM\rangle\langle OM|$, and $U$ represents the payoff under classical game theory.

### 4.2. Analysis of the Quantum Game Model

By analyzing Table 1, it can be seen that under the conditions $a_2 - a_3 < a_5$ and $a_2 - a_3 > a_5 - k$, there is no pure strategy Nash equilibrium solution for this problem in the traditional classical game conditions. The process of government A and automakers B solving the mixed Nash equilibrium in the classical game is shown in Formulas (6)–(11).

$$\pi_A = x[(a_1 - a_6)y + (a_1 + k - a_4 - a_6)(1-y)] + (1-x)[a_1y + (a_1 - a_4)(1-y)] \tag{6}$$

$$\pi_B = y[(a_6 + a_2 - a_3)x + (a_5 + a_6 - k)(1-x)] + (1-y)[(a_2 - a_3)x + a_5(1-x)] \tag{7}$$

$$\pi_A(x^*, y^*) \geq \pi_A(x, y^*) \tag{8}$$

$$\frac{\partial \pi_A}{\partial x} = k - a_6 - ky = 0 \tag{9}$$

$$\pi_B(x^*, y^*) \geq \pi_B(x^*, y) \tag{10}$$

$$\frac{\partial \pi_B}{\partial y} = a_6 - k + kx = 0 \tag{11}$$

The mixed Nash equilibrium solution in the classical game can be derived as follows: $x^* = \frac{k - a_6}{k}$ and $y^* = \frac{k - a_6}{k}$. The payoffs for government A and automakers B in the mixed Nash equilibrium are $\pi_A = -\frac{a_4 a_6 - a_1 k}{k}$ and $\pi_B = \frac{a_3 a_6 - a_2 a_6 + a_5 a_6 + a_2 k - a_3 k}{k}$.

**Proposition 1.** *Quantum game can enhance Nash equilibrium solutions in scenarios compared to classical game, and the outcomes of government A and automakers B, after quantum processing, will be at least as good as the optimal results of classical game strategies.*

**Proof of Proposition 1.** To determine whether the strategy $x^* = 0$ and $y^* = 0$ can achieve an improvement in the Nash equilibrium solution, substituting into Equations (4) and (5), we obtain the following:

$$\pi'_A(0,0) = |a|^2(a_1 - a_4) + |d|^2(a_1 - a_6) + |c|^2(a_1 - a_4 - a_6 + k) + a_1|b|^2 \tag{12}$$

$$\pi'_B(0,0) = |b|^2(a_2 - a_3) + |c|^2(a_5 + a_6 - k) + |d|^2(a_2 - a_3 + a_6) + a_5|a|^2 \tag{13}$$

Transforming the above equation into the optimization problem as in Formula (14), we obtain the following:

$$s.t.\begin{cases} \pi'_A(0,0) \geq -\frac{a_4 a_6 - a_1 k}{k} \\ \pi'_B(0,0) \geq \frac{a_3 a_6 - a_2 a_6 + a_5 a_6 + a_2 k - a_3 k}{k} \end{cases} \tag{14}$$

Parameters $a_1$, $a_2$, $a_3$, $a_4$, $a_5$, $a_6$, and $k$ are all non-negative values. When $c = \frac{a_6}{k}$ and $d = 1 - \frac{a_6}{k}$, they satisfy the constraint on the left side equaling the constraint on the right side. When $a_4 < k$ and $a_2 - a_3 - a_5 + k < 0$, $c = 1$ satisfies the condition of the left side being greater than the right side. When $a_4 > k$ and $a_2 - a_3 - a_5 + k > 0$, $d = 1$ satisfies the condition of the left side being greater than the right side. When $2a_6 > k$ and $a_3 - a_2 + a_5 < 0$, $a = b = \frac{1}{2}$ satisfies the condition of the left side being greater than the right side. When $a_4 + a_6 > k$ and $a_3 - a_2 + a_5 + a_6 - k < 0$, $b = \frac{a_6}{k}$, $d = 1 - \frac{a_6}{k}$ satisfies the condition of the left side being greater than the right side. When $a_4 + a_6 < k$ and $a_3 - a_2 + a_5 + a_6 - k < 0$, $a = \frac{a_6}{k}$, $c = 1 - \frac{a_6}{k}$ satisfies the condition of the left side being greater than the right side. If $x^* = 0$ and $y^* = 0$ is a Nash equilibrium solution, the outcomes after quantum processing will be at least as good as the optimal results of classical game strategies.

To determine whether the strategy $x^* = 1$ and $y^* = 0$ can achieve an improved Nash equilibrium, we substitute into Equations (4) and (5) to obtain the following:

$$\pi'_A(1,0) = (a_1 - a_4 - a_6 + k)|a|^2 + (a_1 - a_6)|b|^2 + (a_1 - a_4)|c|^2 + a_1|d|^2 \tag{15}$$

$$\pi'_B(1,0) = (a_5 + a_6 - k)|a|^2 + (a_2 - a_3 + a_6)|b|^2 + a_5|c|^2 + (a_2 - a_3)|d|^2 \tag{16}$$

Transforming the above equation into the optimization problem as in Formula (17) yields the following:

$$s.t.\begin{cases} \pi'_A(1,0) \geq -\frac{a_4 a_6 - a_1 k}{k} \\ \pi'_B(1,0) \geq \frac{a_3 a_6 - a_2 a_6 + a_5 a_6 + a_2 k - a_3 k}{k} \end{cases} \tag{17}$$

Similar to the proof for $x^* = 0$, $y^* = 0$, it suffices to interchange $a$ and $c$, as well as $b$ and $d$, in the proof process of $x^* = 0$, $y^* = 0$ to obtain the proof process for $x^* = 1$, $y^* = 0$. If $x^* = 1$, $y^* = 0$ is a Nash equilibrium solution, the outcomes after quantum processing will be at least as good as the optimal results of classical game strategies.

To determine whether the strategy $x^* = 0$ $y^* = 1$ can achieve an improved Nash equilibrium solution, substituting into Equations (4) and (5) yields the following:

$$\pi'_A(0,1) = a_1|a|^2 + (a_1 - a_4)|b|^2 + (a_1 - a_6)|c|^2 + (a_1 - a_4 - a_6 + k)|d|^2 \tag{18}$$

$$\pi'_B(0,1) = (a_2 - a_3)|a|^2 + a_5|b|^2 + (a_2 - a_3 + a_6)|c|^2 + (a_5 + a_6 - k)|d|^2 \tag{19}$$

Transforming the above equation into an optimization problem as in Formula (20) obtains the following:

$$s.t.\begin{cases} \pi'_A(0,1) \geq -\frac{a_4 a_6 - a_1 k}{k} \\ \pi'_B(0,1) \geq \frac{a_3 a_6 - a_2 a_6 + a_5 a_6 + a_2 k - a_3 k}{k} \end{cases} \tag{20}$$

Similar to the proof process of $x^* = 0$, $y^* = 0$, by interchanging $a$ with $b$ and $c$ with $d$ in the proof process of $x^* = 0$, $y^* = 0$ to obtain the proof process for $x^* = 0$, $y^* = 1$. If $x^* = 0$, $y^* = 1$ is a Nash equilibrium solution, the outcomes after quantum processing will be at least as good as the optimal results of classical game strategies.

To determine whether the strategy $x^* = 1$, $y^* = 1$ can achieve an improvement in achieving a Nash equilibrium, we can substitute Formulas (4) and (5) to obtain the following:

$$\pi'_A(1,1) = (a_1 - a_6)|a|^2 + (a_1 - a_4 - a_6 + k)|b|^2 + a_1|c|^2 + (a_1 - a_4)|d|^2 \tag{21}$$

$$\pi'_B(1,1) = (a_2 - a_3 + a_6)|a|^2 + (a_5 + a_6 - k)|b|^2 + (a_2 - a_3)|c|^2 + a_5|d|^2 \tag{22}$$

Transforming the above equation into an optimization problem as in Formula (23) yields the following:

$$s.t. \begin{cases} \pi'_A(1,1) \geq -\frac{a_4 a_6 - a_1 k}{k} \\ \pi'_B(1,1) \geq \frac{a_3 a_6 - a_2 a_6 + a_5 a_6 + a_2 k - a_3 k}{k} \end{cases} \tag{23}$$

Similar to the proof for $x^* = 0$, $y^* = 0$, it is sufficient to interchange $a$ and $d$ as well as $b$ and $c$ in the proof for $x^* = 0$, $y^* = 0$ to obtain the proof process for $x^* = 1$, $y^* = 1$. If $x^* = 1$, $y^* = 1$ is a Nash equilibrium solution, the outcomes after quantum processing will be at least as good as the optimal results of classical game strategies. □

Proposition 1 demonstrates that, under certain conditions, quantum processing may alter the dynamics of market games, thereby influencing the long-term strategies of market participants. This can be utilized to analyze the long-term stability and trends in the market [43]. The government can employ quantum processing to assess and select automakers for research, development, production, and innovation, ensuring that resources are allocated to companies with a promising chance of success. Through quantum processing, automakers can better evaluate the impact of different policy options on their risk exposure, enabling them to formulate more effective risk management strategies.

**Proposition 2.** *When the government's fine amount exceeds twice the subsidy amount, i.e., $2a_6 - k < 0$, if the quantum entanglement $|a|^2 + |b|^2 > |c|^2 + |d|^2$, the optimal strategy for the government is to provide subsidizing and penalizing. Conversely, if $|a|^2 + |b|^2 < |c|^2 + |d|^2$, the optimal strategy for the government is not to provide subsidizing and penalizing. When $2a_6 - k > 0$, the above conclusions are reversed.*

**Proposition 3.** *When the total income from the R&D and production of low-carbon transportation tools by the automakers exceeds the total income from the production and sale of traditional transportation tools, i.e., $2a_2 - 2a_3 - 2a_5 + k > 0$, if the quantum entanglement $|a|^2 + |c|^2 > |b|^2 + |d|^2$, the optimal strategy for the automakers is to R&D and production. Conversely, if $|a|^2 + |c|^2 < |b|^2 + |d|^2$, the optimal strategy for the automakers is not to R&D and production. When $2a_2 - 2a_3 - 2a_5 + k < 0$, the above conclusions are reversed.*

**Proof of Proposition 2 and Proposition 3.** Proposition 2 and Proposition 3 prove that by using Equations (4) and (5), we can obtain the following:

$$\begin{aligned} \pi'_A(x^*, y^*) - \pi'_A(x, y^*) &= (x - x^*)(a_6|a|^2 + a_6|b|^2 - a_6|c|^2 - a_6|d|^2 - k|a|^2 \\ &\quad + k|c|^2 + ky^*|a|^2 - ky^*|b|^2 - ky^*|c|^2 + ky^*|d|^2) \end{aligned} \tag{24}$$

$$\begin{aligned} \pi'_B(x^*, y^*) - \pi'_B(x^*, y) &= -(y - y^*)(a_2|a|^2 - a_3|a|^2 - a_5|a|^2 - a_2|b|^2 + a_3|b|^2 \\ &\quad + a_5|b|^2 + a_2|c|^2 - a_3|c|^2 - a_5|c|^2 - a_2|d|^2 + a_3|d|^2 + a_5|d|^2 + k|c|^2 - k|d|^2) \end{aligned} \tag{25}$$

Substituting $x^* = 0, y^* = 0$ strategies into Equations (24) and (25), we obtain the following:

$$\begin{cases} (a_6|a|^2 + a_6|b|^2 - a_6|c|^2 - a_6|d|^2 - k|a|^2 + k|c|^2) \geq 0 \\ -(a_2|a|^2 - a_3|a|^2 - a_5|a|^2 - a_2|b|^2 + a_3|b|^2 + a_5|b|^2 + a_2|c|^2 \\ -a_3|c|^2 - a_5|c|^2 - a_2|d|^2 + a_3|d|^2 + a_5|d|^2 + k|c|^2 - k|d|^2) \geq 0 \end{cases} \quad (26)$$

Substituting $x^* = 1, y^* = 0$ strategies into Equations (24) and (25), we obtain the following:

$$\begin{cases} a_6|a|^2 + a_6|b|^2 - a_6|c|^2 - a_6|d|^2 - k|a|^2 + k|c|^2 \leq 0 \\ a_2|a|^2 - a_3|a|^2 - a_5|a|^2 - a_2|b|^2 + a_3|b|^2 + a_5|b|^2 + a_2|c|^2 \\ -a_3|c|^2 - a_5|c|^2 - a_2|d|^2 + a_3|d|^2 + a_5|d|^2 + k|a|^2 - k|b|^2 \leq 0 \end{cases} \quad (27)$$

Substituting $x^* = 0, y^* = 1$ strategies into Equations (24) and (25), we obtain the following:

$$\begin{cases} a_6|a|^2 + a_6|b|^2 - a_6|c|^2 - a_6|d|^2 - k|b|^2 + k|d|^2 \geq 0 \\ a_2|a|^2 - a_3|a|^2 - a_5|a|^2 - a_2|b|^2 + a_3|b|^2 + a_5|b|^2 + a_2|c|^2 \\ -a_3|c|^2 - a_5|c|^2 - a_2|d|^2 + a_3|d|^2 + a_5|d|^2 + k|c|^2 - k|d|^2 \geq 0 \end{cases} \quad (28)$$

Substituting $x^* = 1, y^* = 1$ strategies into Equations (24) and (25), we obtain the following:

$$\begin{cases} a_6|a|^2 + a_6|b|^2 - a_6|c|^2 - a_6|d|^2 - k|b|^2 + k|d|^2 \leq 0 \\ a_2|a|^2 - a_3|a|^2 - a_5|a|^2 - a_2|b|^2 + a_3|b|^2 + a_5|b|^2 + a_2|c|^2 \\ -a_3|c|^2 - a_5|c|^2 - a_2|d|^2 + a_3|d|^2 + a_5|d|^2 + k|a|^2 - k|b|^2 \geq 0 \end{cases} \quad (29)$$

When the optimal strategy for the government is to provide subsidizing and penalizing, that is, $x^* = 1$, Equations (26) and (28) do not hold, and we have the following:

$$\begin{cases} a_6|a|^2 + a_6|b|^2 - a_6|c|^2 - a_6|d|^2 - k|b|^2 + k|d|^2 < 0 \\ a_6|a|^2 + a_6|b|^2 - a_6|c|^2 - a_6|d|^2 - k|a|^2 + k|c|^2 < 0 \end{cases} \quad (30)$$

$$\begin{array}{l} 2a_6 - k < 0 \Rightarrow |a|^2 + |b|^2 > |c|^2 + |d|^2 \\ 2a_6 - k > 0 \Rightarrow |a|^2 + |b|^2 < |c|^2 + |d|^2 \end{array} \quad (31)$$

When the optimal strategy for the government is not to provide subsidizing and penalizing, that is, $x^* = 0$, Equations (27) and (29) do not hold, and we have:

$$\begin{cases} a_6|a|^2 + a_6|b|^2 - a_6|c|^2 - a_6|d|^2 - k|a|^2 + k|c|^2 > 0 \\ a_6|a|^2 + a_6|b|^2 - a_6|c|^2 - a_6|d|^2 - k|b|^2 + k|d|^2 > 0 \end{cases} \quad (32)$$

$$\begin{array}{l} 2a_6 - k < 0 \Rightarrow |a|^2 + |b|^2 < |c|^2 + |d|^2 \\ 2a_6 - k > 0 \Rightarrow |a|^2 + |b|^2 > |c|^2 + |d|^2 \end{array} \quad (33)$$

When the optimal strategy for the automakers is to engage in R&D and produce, i.e., $y^* = 1$, Equations (26) and (27) do not hold, we have the following:

$$\begin{cases} a_2|a|^2 - a_3|a|^2 - a_5|a|^2 - a_2|b|^2 + a_3|b|^2 + a_5|b|^2 + a_2|c|^2 \\ -a_3|c|^2 - a_5|c|^2 - a_2|d|^2 + a_3|d|^2 + a_5|d|^2 + k|a|^2 - k|b|^2 > 0 \\ -(a_2|a|^2 - a_3|a|^2 - a_5|a|^2 - a_2|b|^2 + a_3|b|^2 + a_5|b|^2 + a_2|c|^2 \\ -a_3|c|^2 - a_5|c|^2 - a_2|d|^2 + a_3|d|^2 + a_5|d|^2 + k|c|^2 - k|d|^2) < 0 \end{cases} \quad (34)$$

$$2a_2 - 2a_3 - 2a_5 + k > 0 \Rightarrow |a|^2 + |c|^2 > |b|^2 + |d|^2$$
$$2a_2 - 2a_3 - 2a_5 + k < 0 \Rightarrow |a|^2 + |c|^2 < |b|^2 + |d|^2 \tag{35}$$

When the optimal strategy for the automakers is not to engage in R&D and produce, i.e., $y^* = 0$, Equations (28) and (29) do not hold, we have the following:

$$\begin{cases} a_2|a|^2 - a_3|a|^2 - a_5|a|^2 - a_2|b|^2 + a_3|b|^2 + a_5|b|^2 + a_2|c|^2 \\ -a_3|c|^2 - a_5|c|^2 - a_2|d|^2 + a_3|d|^2 + a_5|d|^2 + k|a|^2 - k|b|^2 < 0 \\ a_2|a|^2 - a_3|a|^2 - a_5|a|^2 - a_2|b|^2 + a_3|b|^2 + a_5|b|^2 + a_2|c|^2 \\ -a_3|c|^2 - a_5|c|^2 - a_2|d|^2 + a_3|d|^2 + a_5|d|^2 + k|c|^2 - k|d|^2 < 0 \end{cases} \tag{36}$$

$$2a_2 - 2a_3 - 2a_5 + k > 0 \Rightarrow |a|^2 + |c|^2 < |b|^2 + |d|^2$$
$$2a_2 - 2a_3 - 2a_5 + k < 0 \Rightarrow |a|^2 + |c|^2 > |b|^2 + |d|^2 \tag{37}$$

By Equations (31), (33), (35), and (37), it is evident that Propositions 2 and 3 hold, concluding the proof. □

Proposition 2 suggests that the government can adopt the difference in profits between strategy *V* and strategy *O* as the final decision criterion. When $2a_6 - k < 0$, if $|a|^2 + |b|^2 > |c|^2 + |d|^2$, the government's optimal strategy is to subsidize and penalize; the government should lean towards the quantum entanglement under the subsidizing and penalizing strategy to gain more revenue. If $|a|^2 + |b|^2 < |c|^2 + |d|^2$, the government's optimal strategy is not to subsidize and penalize, and the government should lean towards the quantum entanglement under the no-subsidy and no-penalty strategy to gain more revenue. When $2a_6 - k > 0$, if $|a|^2 + |b|^2 < |c|^2 + |d|^2$, the government's optimal strategy is to subsidize and penalize; the government should lean towards the quantum entanglement under the no-subsidy and no-penalty strategy to gain more revenue. If $|a|^2 + |b|^2 > |c|^2 + |d|^2$, the government's optimal strategy is not to subsidize and penalize, and the government should lean towards the quantum entanglement under the subsidize and penalty strategy to gain more revenue. Quantum entanglement measures the degree to which strategy payoffs are entangled [44]. Under this condition $2a_6 - k > 0$, the government's optimal strategy is not stable, and the government should establish rules where the penalty amount is greater than double the subsidy amount.

In the policy drive to promote the R&D and production of low-carbon transportation vehicles, it is imperative to carefully balance the interplay between economic costs, environmental conservation, and societal well-being. While the R&D and production of low-carbon transportation vehicles hold significant importance in reducing carbon emissions and addressing climate change, we cannot overlook the potential fiscal pressures they may bring about as well as their impacts on other critical sectors. Guiding market development, fostering innovative technologies, and facilitating industrial upgrades constitute sustainable pathways for advancing the development of low-carbon automobiles.

Proposition 3 indicates that automakers can use the difference in profits between strategy *N* and strategy *M* as the final decision criterion. When $2a_2 - 2a_3 - 2a_5 + k > 0$, If $|a|^2 + |c|^2 > |b|^2 + |d|^2$, the optimal strategy for the automakers is to engage in R&D and produce. The automakers should lean towards the quantum entanglement associated with the R&D and produce strategy to maximize their revenue. If $|a|^2 + |c|^2 < |b|^2 + |d|^2$, the optimal strategy for the automakers is not to engage in R&D and production, and in this case, they should lean towards the quantum entanglement associated with the strategy of not engaging in R&D and production to maximize their revenue. When $2a_2 - 2a_3 - 2a_5 + k < 0$, if $|a|^2 + |c|^2 < |b|^2 + |d|^2$, the optimal strategy for the automakers is to engage in R&D and to produce. The automakers should lean towards the quantum entanglement associated with not engaging in R&D and production to maximize their revenue. If $|a|^2 + |c|^2 > |b|^2 + |d|^2$, the optimal strategy for the automakers is not to engage in R&D and production, and in this case, they should lean towards the quantum

entanglement associated with the R&D and production strategy to maximize their revenue. Under the condition $2a_2 - 2a_3 - 2a_5 + k < 0$, the optimal strategy for the automakers is not stable, and the automakers should implement a series of measures to reduce the cost of developing low-carbon transportation and improve production and operational efficiency.

Furthermore, when $|a|^2 + |b|^2 > |c|^2 + |d|^2$, this corresponds to the region enclosed by ABCD in Figure 2, it indicates a higher entanglement between the entangled states $|VN\rangle$ and $|VM\rangle$. When $|a|^2 + |b|^2 < |c|^2 + |d|^2$ corresponding to the region enclosed by GEFH in Figure 2, it suggests a higher entanglement between the entangled states $|ON\rangle$ and $|OM\rangle$. When $|a|^2 + |c|^2 > |b|^2 + |d|^2$ within the region enclosed by CGHD in Figure 2, it signifies a higher entanglement between the entangled states $|VN\rangle$ and $|ON\rangle$. When $|a|^2 + |c|^2 < |b|^2 + |d|^2$ corresponding to the region enclosed by AFEB in Figure 2, it implies a higher entanglement between the entangled states $|VM\rangle$ and $|OM\rangle$.

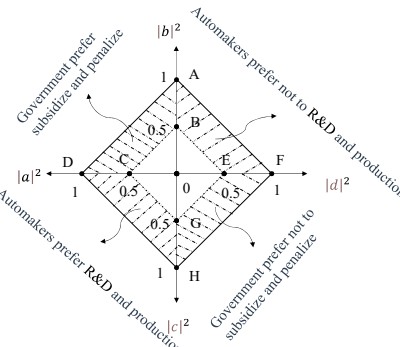

**Figure 2.** Entanglement diagram of different regions.

In summary, the region enclosed by ADHFEGCB is the area we aim to guide. For both automakers and governments, mechanisms can be established to steer the entanglement toward the ADHFEGCB region, thus promoting the development of global low-carbon transportation.

**Proposition 4.** *Government A and automakers B have variable quantum game Nash equilibrium solutions. One of the critical conditions for the solution to change is the initial entanglement, where* $|a|^2 + |d|^2 = |b|^2 + |c|^2 = \frac{1}{2}$.

**Proof of Proposition 4.** Let $|a|^2 = |b|^2 = |c|^2 = |d|^2 = \frac{1}{4}$. Let us substitute these values into Formulas (4) and (5); the payoffs for the government and automakers are shown in Table 2.

**Table 2.** Payoff matrix under the condition $|a|^2 = |b|^2 = |c|^2 = |d|^2 = \frac{1}{4}$.

| Payoffs | | B | |
|---|---|---|---|
| | | *N* | *M* |
| A | *V* | $(a_1 - \frac{a_4}{2} - \frac{a_6}{2} + \frac{k}{4}, \frac{a_2}{2} - \frac{a_3}{2} + \frac{a_5}{2})$ | $(a_1 - \frac{a_4}{2} - \frac{a_6}{2} + \frac{k}{4}, \frac{a_2}{2} - \frac{a_3}{2} + \frac{a_5}{2})$ |
| | *O* | $(a_1 - \frac{a_4}{2} - \frac{a_6}{2} + \frac{k}{4}, \frac{a_2}{2} - \frac{a_3}{2} + \frac{a_5}{2})$ | $(a_1 - \frac{a_4}{2} - \frac{a_6}{2} + \frac{k}{4}, \frac{a_2}{2} - \frac{a_3}{2} + \frac{a_5}{2})$ |

By analyzing Table 2, it can be observed that when $a_2 - a_3 < a_5$ and $4a_1 - 2a_4 - 2a_6 + k < 0$, the Nash equilibrium solutions are $(V, N)$, $(V, M)$, $(O, N)$, and $(O, M)$.

Now, let us discuss the variation in Nash equilibrium solutions. We change the condition from $|a|^2 = |b|^2 = |c|^2 = |d|^2 = \frac{1}{4}$ to $|a|^2 = \frac{3}{8}, |b|^2 = |c|^2 = \frac{1}{4}, |d|^2 = \frac{1}{8}$. Substituting these values into Formulas (4) and (5), we obtain the government's and the automakers' profits as shown in Table 3.

**Table 3.** Payoff matrix under the condition $|a|^2 = \frac{3}{8}, |b|^2 = |c|^2 = \frac{1}{4}, |d|^2 = \frac{1}{8}$.

| Payoffs | | B | |
|---|---|---|---|
| | | N | M |
| A | V | $(a_1 - \frac{3a_4}{8} - \frac{5a_6}{8} + \frac{k}{4},$ $\frac{5a_2}{8} - \frac{5a_3}{8} + \frac{3a_5}{8} + \frac{5a_6}{8} - \frac{k}{4})$ | $(a_1 - \frac{5a_4}{8} - \frac{5a_6}{8} + \frac{3k}{8},$ $\frac{3a_2}{8} - \frac{3a_3}{8} + \frac{5a_5}{8} + \frac{5a_6}{8} - \frac{3k}{8})$ |
| | O | $(a_1 - \frac{3a_4}{8} - \frac{3a_6}{8} + \frac{k}{8},$ $\frac{5a_2}{8} - \frac{5a_3}{8} + \frac{3a_5}{8} + \frac{3a_6}{8} - \frac{k}{8})$ | $(a_1 - \frac{5a_4}{8} - \frac{3a_6}{8} + \frac{k}{4},$ $\frac{3a_2}{8} - \frac{3a_3}{8} + \frac{5a_5}{8} + \frac{3a_6}{8} - \frac{k}{4})$ |

Based on Table 3, we can observe that when $k > 2a_6$ and $2a_2 - 2a_3 - 2a_5 + k > 0$, the Nash equilibrium solution is ($V$,$N$). When $k < 2a_6$ and $2a_2 - 2a_3 - 2a_5 + k < 0$, the Nash equilibrium solution becomes ($O$,$N$). In the case where $2a_2 - 2a_3 - 2a_5 + k < 0$, the Nash equilibrium solution changes to ($V$,$M$). Similarly, when $2a_2 - 2a_3 - 2a_5 + k < 0$, the Nash equilibrium solution shifts to ($O$,$M$).

Changing the conditions once again $|a|^2 = \frac{1}{4}, |b|^2 = \frac{3}{8}, |c|^2 = \frac{1}{8}, |d|^2 = \frac{1}{4}$, and substituting these values into Equations (4) and (5), the earnings are shown in Table 4.

**Table 4.** Payoff matrix under the condition $|a|^2 = \frac{1}{4}, |b|^2 = \frac{3}{8}, |c|^2 = \frac{1}{8}, |d|^2 = \frac{1}{4}$.

| Payoffs | | B | |
|---|---|---|---|
| | | N | M |
| A | V | $(a_1 - \frac{5a_4}{8} - \frac{5a_6}{8} + \frac{3k}{8},$ $\frac{3a_2}{8} - \frac{3a_3}{8} + \frac{5a_5}{8} + \frac{5a_6}{8} - \frac{3k}{8})$ | $(a_1 - \frac{3a_4}{8} - \frac{5a_6}{8} + \frac{k}{4},$ $\frac{5a_2}{8} - \frac{5a_3}{8} + \frac{3a_5}{8} + \frac{5a_6}{8} - \frac{k}{4})$ |
| | O | $(a_1 - \frac{5a_4}{8} - \frac{3a_6}{8} + \frac{k}{4},$ $\frac{3a_2}{8} - \frac{3a_3}{8} + \frac{5a_5}{8} + \frac{3a_6}{8} - \frac{k}{4})$ | $(a_1 - \frac{3a_4}{8} - \frac{3a_6}{8} + \frac{k}{8},$ $\frac{5a_2}{8} - \frac{5a_3}{8} + \frac{3a_5}{8} + \frac{3a_6}{8} - \frac{k}{8})$ |

From Table 4, it can be observed that when $k > 2a_6$ and $-2a_2 + 2a_3 + 2a_5 - k > 0$, the Nash equilibrium is ($V$, $N$). When $k > 2a_6$ and $-2a_2 + 2a_3 + 2a_5 - k < 0$, the Nash equilibrium changes to ($V$, $M$). When $k < 2a_6$ and $-2a_2 + 2a_3 + 2a_5 - k > 0$, the Nash equilibrium is ($O$, $N$). When $k < 2a_6$ and $-2a_2 + 2a_3 + 2a_5 - k < 0$, the Nash equilibrium changes to ($O$, $M$). Proposition 4 is thus proven. □

Proposition 4 states the following: Under the combined effect of government subsidizing and penalizing low-carbon transportation, there is no Nash equilibrium solution in classical game theory. However, the problem can be resolved by applying quantum game theory, where changes in entanglement affect the game outcomes and Nash equilibrium solutions. Under the conditions $2a_6 - k < 0$ and $2a_2 - 2a_3 - 2a_5 + k < 0$, both the government and automakers can adjust entanglement by choosing a quantum bit identity operator $E$ and quantum bit flip operator $C$ to satisfy the conditions $|a|^2 + |b|^2 > |c|^2 + |d|^2$ and $|a|^2 + |c|^2 < |b|^2 + |d|^2$, thus maximizing profits. This enables the transition from other quantum strategies to the ($V$, $N$) quantum strategy, ensuring mutual efforts by the government and automakers in the direction of low-carbon transportation development and creating more social value.

## 5. The Impact of Quantum Entanglement on Simulation Analysis under Quantum State Superposition

Due to the bounded rationality of both government A and automakers B, the game between these two entities cannot instantaneously reach a quantum Nash equilibrium. Instead, it requires a gradual emergence of quantum superposition states through repeated gameplay to attain a quantum equilibrium state. The impact of the initial entangled state in a quantum superposition state on the profits of government A and automakers B is significant, necessitating a numerical simulation analysis.

In the presence of stochastic or uncertain factors, Monte Carlo simulation analysis is of remarkable significance. Therefore, Monte Carlo simulation analysis is employed, generating 100,000 random samples using matlab. The parameter values chosen in our research were based on a combination of theoretical considerations and empirical evidence from previous studies in the field [20,38–40]. Additionally, we consulted with experts in the field to ensure that our chosen parameter values were reasonable and aligned with existing knowledge in the area. Without loss of generality, the parameters $a_2$, $a_5$, $a_6$ are rounded, let $a_1 = 18$, $a_4 = 5$, $a_2 = 34$, $a_3 = 25$, $a_5 = 7$, $a_6 = 2$, and $k = 8$. The unit is RMB 10,000 per vehicle. For the accuracy and reproducibility of simulation results, let $|a|^2 = \left(\cos\frac{\theta_M}{2}\cos\frac{\theta_F}{2}\right)^2$, $|b|^2 = \left(\cos\frac{\theta_M}{2}\sin\frac{\theta_F}{2}\right)^2$, $|c|^2 = \left(\sin\frac{\theta_M}{2}\cos\frac{\theta_F}{2}\right)^2$, and $|d|^2 = \left(\sin\frac{\theta_M}{2}\sin\frac{\theta_F}{2}\right)^2$ [25].

It can be obtained that government A's quantum game payoff compared to the classical game payoff under the strategy $|VN\rangle$ is illustrated in Figure 3. If $|VN\rangle$ represents a Nash equilibrium solution and the payoff under the $|VN\rangle$ strategy is positive, then the quantum game's payoff is superior to the optimal classical game strategy payoff. This outcome aligns with Proposition 1. When different initial entangled states are considered, it is observed that the payoff under the $|VN\rangle$ strategy becomes negative. In such cases, the Nash equilibrium solution changes, leading to the adoption of one of the other three strategies, which is consistent with Proposition 4. Due to space limitations, the graphs showing the difference between government A's quantum game payoff and the classical game payoff under the other three strategies are not provided here. Given that government A adopts a quantum strategy with a higher payoff compared to the traditional classical strategy, there is no incentive for government A to adopt the classical strategy. Government A can use quantum strategies to assess and select automakers for research, development, production, and innovation to maximize its returns.

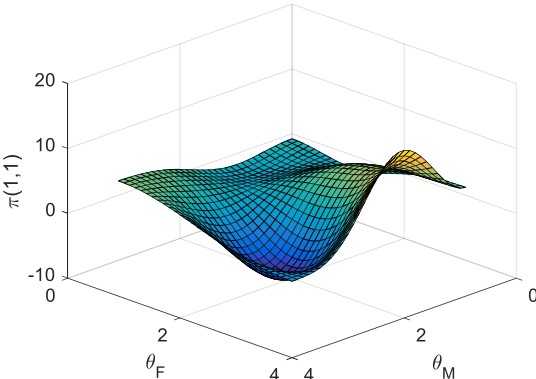

**Figure 3.** The difference in government A quantum game payoff compared to the optimal classical game payoff under $|VN\rangle$ strategy.

In Figure 4, the quantum game payoff for automakers B under the $|VN\rangle$ strategy compared to the classical game payoff is shown. When $|VN\rangle$ represents a Nash equilibrium solution and the payoff under the $|VN\rangle$ strategy is positive, then the quantum game's payoff is superior to the optimal classical game strategy payoff. This result aligns with Proposition 1. As automakers B adopt a quantum game strategy with a higher payoff compared to the classical game strategy, there is no motivation for automakers B to opt for the classical strategy. Automakers B can use quantum strategies to better assess the impact of the government's different subsidy and penalty policies on their returns, thus maximizing their payoff.

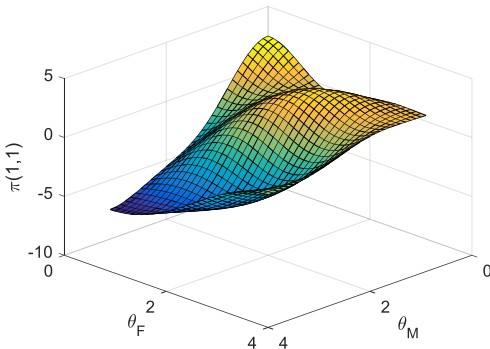

**Figure 4.** The difference in automakers B quantum game payoff compared to the optimal classical game payoff under $|VN\rangle$ strategy.

Under the condition $|a|^2 + |b|^2 > |c|^2 + |d|^2$, the government can use the difference in profits between strategy $V$ and strategy $O$ as the final decision criterion, as shown in Figure 5. It is observed that government A's strategy $V$ is significantly superior to strategy $O$, in line with the conclusion of Proposition 2. Increasing the values of the initial entangled states $|a|^2$ and $|b|^2$ is favorable for government A's strategy $V$ to achieve higher profits. Government A can enhance its revenue by increasing the value of $|a|^2$ and reducing subsidy measures. Government A can also increase the value of $|b|^2$ and decrease subsidies while increasing penalty measures to improve their revenue.

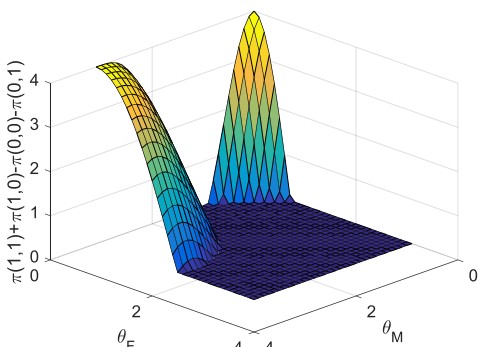

**Figure 5.** The difference in payoff between $V$ strategy and $O$ strategy under the condition $|a|^2 + |b|^2 > |c|^2 + |d|^2$.

Similarly, under the condition $|a|^2 + |b|^2 < |c|^2 + |d|^2$, as depicted in Figure 6, it is found that government A's strategy $O$ is significantly better than strategy $V$, aligning with the conclusion of Proposition 2. Increasing the values of the initial entangled states $|c|^2$ and $|d|^2$ is advantageous for government A's strategy $O$ to attain higher profits. Government A can improve its revenue by increasing the value of $|c|^2$ and enhancing tax measures. Government A can also increase the value of $|d|^2$ and raise tax measures while reducing environmental loss measures to boost their revenue.

Under the condition $|a|^2 + |c|^2 > |b|^2 + |d|^2$, automakers B can use the difference in profits between strategy $N$ and strategy $M$ as the final decision criterion, as shown in Figure 7. It is observed that automakers B's strategy $N$ is significantly superior to strategy $M$, in alignment with the conclusion of Proposition 3. Increasing the values of the initial entangled states $|a|^2$ and $|c|^2$ is advantageous for automakers B's strategy "N" to achieve higher profits. Automakers B can enhance their revenue by increasing the value of $|a|^2$ and implementing cost-effective measures. However, automakers B should not increase the value of $|c|^2$, as this may lead automakers B to lose interest in R&D and producing low-carbon transportation solutions. To prevent this outcome, the government needs to reduce subsidies and increase penalties.

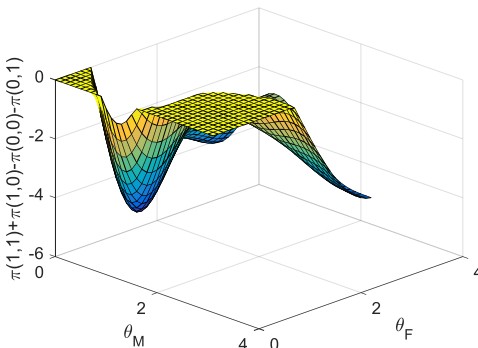

**Figure 6.** The difference in payoff between *V* strategy and *O* strategy under the condition $|a|^2 + |b|^2 < |c|^2 + |d|^2$.

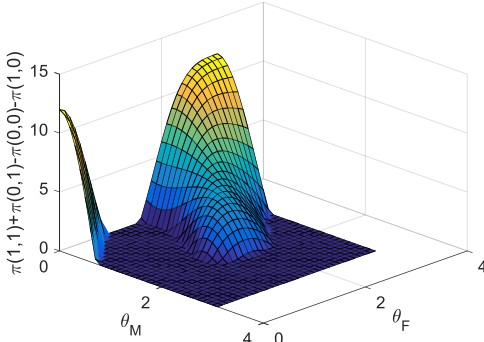

**Figure 7.** The difference in payoff between *N* strategy and *M* strategy under the condition $|a|^2 + |c|^2 > |b|^2 + |d|^2$.

Likewise, under the condition $|a|^2 + |c|^2 < |b|^2 + |d|^2$, as illustrated in Figure 8, it is found that automakers B's strategy "M" is significantly better than strategy "N", in agreement with the conclusion of Proposition 3. Increasing the values of the initial entangled states $|b|^2$ and $|d|^2$ is beneficial for automakers B's strategy "M" to achieve higher profits. Automakers B can improve their revenue by increasing the value of $|b|^2$ and implementing cost-effective measures. However, automakers B should not increase the value of $|d|^2$, as this may lead automakers B to lose interest in R&D and producing low-carbon transportation solutions. In order to prevent automakers from losing interest in the development and production of low-carbon transportation vehicles, the government needs to implement subsidy and penalty policies to incentivize automakers to engage in the R&D and production of low-carbon transportation vehicles.

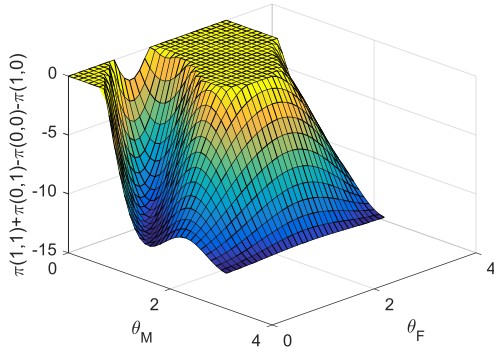

**Figure 8.** The difference in payoff between *N* strategy and *M* strategy under the condition $|a|^2 + |c|^2 < |b|^2 + |d|^2$.

### 6. Summary and Insights

Low-carbon transportation has become increasingly crucial for global transportation development. In particular, the research and production of low-carbon transportation solutions by automakers are of paramount importance. Government subsidies and penalties play a significant role in incentivizing automakers to develop and manufacture low-carbon transportation tools. However, traditional classical game theory often falls short in providing effective incentives. This study considers the joint effects of quantum initial entanglement states and quantum superposition states in quantum games on the government's low-carbon subsidies and penalties for automakers. The research outcomes are expected to drive the global low-carbon transportation sector, accelerating the transformation and development of the global low-carbon economy. The primary conclusions are as follows:

(1) Quantum initial entanglement states and quantum superposition states have a vital influence on the low-carbon strategies of both the government and automakers. These states are also crucial factors affecting the revenue of both parties. There is a substantial synergy between the government's low-carbon subsidies and penalties and quantum initial entanglement states and quantum superposition states. The government can enhance revenue significantly by steering quantum initial entanglement states to align with areas where automakers engage in research and development and by adopting quantum strategies that lead to increased profits for both the government and automakers.

(2) Quantum processing can create new gaming strategies that are more conducive to achieving Nash equilibria and improved Nash equilibrium solutions. It also facilitates the implementation of stable strategies, with the optimal results at least matching, if not surpassing, those of classical game theory. Governments can use quantum game theory to formulate policies that effectively promote the development of low-carbon transportation while minimizing environmental impact.

(3) The amount of government penalties should exceed double the number of subsidies, and government policies should ensure that automakers' profits from developing and manufacturing low-carbon transportation solutions, under subsidy conditions, surpass those from producing and selling traditional transportation tools. Automakers should take a series of measures to reduce production costs, enhance production efficiency, and improve operational performance.

(4) In quantum games, automakers' preference for not R&D and producing low-carbon transportation solutions, as reflected by quantum entanglement behavior, significantly increases. The government should regulate and mitigate automakers' preference for avoiding the development of low-carbon transportation tools through quantum entanglement. By exploiting quantum entanglement phenomena, the government and automakers can further increase the correlation of low-carbon transportation revenues, thereby promoting the growth of the low-carbon transportation market.

In comparison with other studies, our findings are consistent with previous research in suggesting that quantum game yields generally outperform classical game yields, or at least do not underperform [18,19,25,26]. However, some differences exist, particularly in research methodologies [10–17] and the treatment of research subjects [27–36]. Our unique Propositions 2, 3, and 4 facilitate achieving Nash equilibrium and perfect coordination in quantum games, significantly enhancing the yields for both parties.

Based on the aforementioned research, the following insights are provided:

(1) The quantum game theory provides a novel framework for governments to guide automakers in transitioning towards low-carbon transportation. By formulating rational subsidy and penalty policies, governments can influence the profitability of both themselves and automakers. This approach fosters an ecosystem where quantum game technology enhances decision-making, ensuring informed policy formulation and optimized outcomes.

(2)  The critical parameter of quantum initial entanglement states profoundly impacts the revenues of both governments and automakers. Governments should guide these states to align with areas of research and production, incentivizing automakers to actively participate in low-carbon transportation innovation. Quantum state superposition enables cooperative and non-cooperative behavior, urging decisions that balance effectiveness with collaboration potential.

(3)  Establishing a quantum game regulatory framework is paramount to prevent automakers from misusing quantum states to evade low-carbon requirements. Automakers can leverage quantum game analysis to collaborate with governments, maximizing the benefits of subsidies and penalties. Emphasizing cost reduction, efficiency improvement, and product quality ensures that revenue from low-carbon transportation surpasses traditional methods, driving sustainable development.

(4)  Under the joint effects of government subsidies and penalties, various considerations exist. Firstly, as demonstrated in this study, the government provides subsidies to automakers who engage in low-carbon research and production, without imposing penalties. In cases where automakers fail to develop low-carbon transportation solutions, the government applies penalties. Secondly, building upon this study, when automakers invest in low-carbon transportation solutions, the government provides further incentives. Lastly, when automakers invest in low-carbon transportation solutions, the government offers subsidies. However, if automakers fail to do so, the government imposes penalties.

## 7. Future Prospects

The study of quantum games holds significant guiding implications for promoting governmental subsidies and penalties in the development of low-carbon transportation. However, the article also exhibits several shortcomings:

(1)  The section on model establishment employs a simplified form, with the analysis process being an aggregate model, failing to conduct analyses from a local perspective.
(2)  The numerical simulation analysis part is limited, as random outcomes fail to gather and analyze real-world data.
(3)  This research primarily focuses on the theoretical aspects of quantum game theory, addressing problems through theoretical deduction and analysis, thus possessing certain limitations in practical application.
(4)  Considering the limitations in the practical application phase, the conclusions drawn in this article may have far-reaching impacts. Looking ahead, my objective is to address these limitations and explore avenues to broaden the practical applicability of the research.

The realization of quantum mechanics demands a highly controlled environment, leading to numerous challenges in applying quantum game theory to practical scenarios. With the continuous success of quantum experiments, it is believed that the technological realization of quantum games will soon be achieved [27,28]. Subsequent research will take into account the aforementioned shortcomings.

**Author Contributions:** Conceptualization, Y.L. and J.W.; methodology, J.W.; software, J.W.; validation, Y.L.; formal analysis, B.W.; investigation, B.W.; resources, B.W.; data curation, Y.L.; writing—original draft preparation, J.W.; writing—review and editing, B.W.; visualization, C.L.; supervision, C.L.; project administration, C.L.; funding acquisition, Y.L. All authors have read and agreed to the published version of the manuscript.

**Funding:** This research has received funding from the National Social Science Fund project (22XTQ008), the Soft Science Project of the Shaanxi Science and Technology Department (2024ZC-YBXM-067), Special Think Tank Project of Shaanxi Philosophy and Social Science Research (2024ZD483).

**Institutional Review Board Statement:** Not applicable.

**Informed Consent Statement:** Not applicable.

**Data Availability Statement:** Data are contained within the article.

**Acknowledgments:** We sincerely express our gratitude to the editor and anonymous referees for their valuable comments and suggestions.

**Conflicts of Interest:** The authors declare no conflicts of interest.

## Abbreviations

| | |
|---|---|
| A | Government |
| B | Automakers |
| R&D | Research and development |
| *V* | Subsidize and penalize |
| *O* | Not subsidize and not penalize |
| *N* | R&D and production |
| *M* | Not to R&D and production |

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
