# Peer review of "A Study of Quantum Game for Low-Carbon Transportation with Government Subsidies and Penalties"

_sustainability, doi:10.3390/su16073051_

Round 1

Reviewer 1 Report

Comments and Suggestions for Authors

This study employs quantum game theory and Nash equilibrium theory to investigate the quantum game problem influenced by government subsidies and penalties for low-carbon transportation.  Your research offers an intriguing perspective to address the low-carbon transportation issue, holding significant theoretical and practical value. After a thorough review of your paper, I believe there are several key areas that require further revisions and enhancements. My suggestions are as follows:

  1. 1. Addition of an Abbreviation List: To clarify the content, I recommend adding an abbreviation list at the beginning of the article. This would greatly assist readers in quickly understanding and referencing the frequent technical terms and abbreviations used throughout your paper.

  2. 2. Quantum Game Theory Calculation Process and Fuzzy Decision-Making: Introducing fuzzy decision-making into the quantum game theory calculation process in your paper is an innovative attempt, adding novelty to your study. However, simply incorporating a probability variable “x” to represent this fuzziness may not fully convey the methodological innovation. I suggest further exploring and discussing how to model the uncertainty between corporate decisions and government subsidies in a more in-depth and complex manner. For instance, consider employing empirical data-based fuzzy logic or other advanced uncertainty modeling methods to more comprehensively capture this complexity.

  3. 3. Lack of Empirical Data: While a computational model is proposed, the absence of empirical data support could weaken the paper's persuasive power. To better showcase industry changes and validate the model's effectiveness and applicability, I advise collecting and analyzing actual data. Specifically, your model should be validated by examining the relationship between actual subsidies and low-carbon emission vehicle types in the Chinese automotive market. This would not only strengthen the empirical foundation of your paper but also provide readers with more convincing case studies.

In summary, your research makes important theoretical and methodological contributions to exploring the quantum game issue in low-carbon transportation. I am confident that by carefully considering and revising according to the above suggestions, your paper will achieve further improvement in quality and impact.

Comments on the Quality of English Language

1. Ensure consistent use of terms throughout the paper. For example, if "low-carbon transportation" and "sustainable transportation" are used interchangeably, define them early in the paper to avoid confusion.

2. Make sure that all terms and concepts are defined clearly when first introduced. Additionally, avoid vague language and be as precise as possible in descriptions and arguments.

Author Response

Comment 1: Addition of an Abbreviation List: To clarify the content, I recommend adding an abbreviation list at the beginning of the article. This would greatly assist readers in quickly understanding and referencing the frequent technical terms and abbreviations used throughout your paper.

Response: An abbreviation list has been incorporated into the original text as per your request.

Comment 2: Quantum Game Theory Calculation Process and Fuzzy Decision-Making: Introducing fuzzy decision-making into the quantum game theory calculation process in your paper is an innovative attempt, adding novelty to your study. However, simply incorporating a probability variable “x” to represent this fuzziness may not fully convey the methodological innovation. I suggest further exploring and discussing how to model the uncertainty between corporate decisions and government subsidies in a more in-depth and complex manner. For instance, consider employing empirical data-based fuzzy logic or other advanced uncertainty modeling methods to more comprehensively capture this complexity.

Response: Fuzzy decision-making methods are presently a straightforward mainstream approach. In the future, we can endeavor to utilize existing resources and tools to learn these methods or collaborate with peers in the field to explore how to apply more complex uncertainty modeling techniques to our research. Gradually, we can enhance our research level to provide more profound and comprehensive solutions for modeling uncertainty between business decisions and government subsidies.

Comment 3: Lack of Empirical Data: While a computational model is proposed, the absence of empirical data support could weaken the paper's persuasive power. To better showcase industry changes and validate the model's effectiveness and applicability, I advise collecting and analyzing actual data. Specifically, your model should be validated by examining the relationship between actual subsidies and low-carbon emission vehicle types in the Chinese automotive market. This would not only strengthen the empirical foundation of your paper but also provide readers with more convincing case studies.

Response: Your point about the importance of empirical data support is highly insightful, and we fully agree with it. Given our current time constraints, we are unable to collect and analyze actual data in the short term. However, we have acknowledged this limitation and plan to dedicate efforts to gather and analyze real-world data from the Chinese automotive market in future research, to validate the effectiveness and applicability of our proposed computational model. We sincerely apologize for being unable to complete this work within the existing revision deadline.

Comment 4: 1. Ensure consistent use of terms throughout the paper. For example, if "low-carbon transportation" and "sustainable transportation" are used interchangeably, define them early in the paper to avoid confusion.2. Make sure that all terms and concepts are defined clearly when first introduced. Additionally, avoid vague language and be as precise as possible in descriptions and arguments.

Response: The language has been polished and modified.

Reviewer 2 Report

Comments and Suggestions for Authors

The paper is very interesting theoretically, the formalism and mathematical argument are clear and readable. However, the paper does not correspond to real-life situations (research and inference about reality) and is unsuitable for publication in its current form.

It is necessary to decide: (1) whether it should be a theoretical one - an elaboration of classical game theory and its study on more or less real data, and (2) whether it should attempt to define the basis of a macroeconomic theory applicable to the area indicated.

Notwithstanding the above recommendation, the following specific comments are relevant.

(9) The wording needs to be improved linguistically, especially No. 2.

(15) The abstract needs to be structured. The research problem should be clearly described, the specific aim of the research should be indicated (theoretical, practical?), the research method should be described, and the results and the added value of the research (theoretical and practical) should be given.

(16) Has the thesis of insufficient incentives been investigated in the paper? Maybe the problem is not the insufficiency of incentives just misplaced economic objectives by governments?

(25) How are these incentives created?

(26) How does this stimulate car manufacturers?

(33) The phrase 'Government Subsidizing and Penalizing' should be corrected. It is not a keyword.

(36) The introduction should be tidied up. It should contain, first of all, structured and synthetic information about the research problem. The research gap should be identified, and the methods with which other authors solve this research problem should be given. The next step should be a discussion of the assumptions of the method proposal given by the authors.

(41) Are the authors confident in this statement?

(60) What is meant by the expression: 'the game theory of low-carbon transportation'? Is it documented in the literature?

(92) This section should be tidied up. Terms taken from physics should be strictly defined. How these concepts are introduced into game theory should be stated. Game theory should be defined with concepts drawn from physics. List what changes the authors propose to this theory.

(110) This section of the paper should be very precisely described.

(111) This section is very general. The facts of the transport sector's contribution to CO2 emissions should be discussed, the policy goals for reducing transport emissions should be precisely characterized, their feasibility should be assessed, and the methods by which this can be achieved should be indicated.

(178) This claim is only true at the scale of quantum objects.

(184) This is an incomprehensible statement. Please clarify it.

(186) This section should be given a scientific research character.

(219) These are only examples, not verified scientific theses. They should be scientifically documented or removed from the paper.

(232) The assumptions used to develop the model should be carefully verified.

1. The functioning of the economy. Theoretically, the strategic choices of government and automakers could be different - four combinations of subsidization and punishment on the government side, and four combinations of R&D and production on the producers' side. In fact, from observing government regulation and the activities of car manufacturers, it appears that governments will simultaneously use subsidies and penalties, and manufacturers already simultaneously use R&D and production (all happening to varying degrees). There are no states to the contrary. It is also a very big oversimplification to aggregate one figure for each of the issues (subsidies, penalties, R&D, production). On the producers' side, shouldn't only profit (which takes into account car sales and not production and R&D expenses separately) be included? All this leads to the conclusion that the model allows for states that do not occur and at the same time is very simplistic. What is the practical utility of such a model? In this context, should the further considerations in the paper be regarded as purely theoretical?

2. Use of quantum theory. When the U process acts, the system is in a superposition of all quantum states. If the initial state and the wave function are known, the probabilities of subsequent states and the further evolution of the system (in superposition) can be calculated. Is this the case under consideration? If so, what are the implications of this fact for the use of the model? The system will assume a particular state when the R process acts and the wave function collapses. Please explain, according to your discussion, what this means for the proposed model. 

(402-405) In this paragraph, conclusions for specific manufacturers are indicated. This is not legitimate in the context of Proposal 1 and previous considerations, which are only concerned with the overall aggregate model. Manufacturers compete with each other and have different business strategies. This needs to be clarified.

(452) The definition of the government's strategy, in the context of Proposition 2, is wrong. The government's objective is not to gain more financial benefit from the strategy, but to achieve the objectives of the stated economic strategy (implicitly: to force manufacturers to produce low-carbon cars, regardless of the economic cost). This eliminates the practical utility of some of the cases analyzed. Please explain.

(452-466), (467-482) The paragraphs contain contradictory conclusions.

(544) Justification should be given for the parameter values adopted. A mere reference to the literature is not sufficient.

(547) This issue, due to its complexity, could serve as the subject of a separate paper. The description is inadequate. What random number generator was used? What was the number of samples? Are the random results consistent with observations of reality?

(608) This claim is completely wrong - where there are subsidies, there is no market.

(613) The conclusions need to be rewritten, taking into account the assumptions made, the extent of the research conducted, and the usefulness of the results. Their current general form is not justified by the considerations presented in the paper.

1. The paper is theoretical. The assumptions made are both simplistic and extensional to the observed practical states. They do not reflect the actual situation concerning the actions of governments and the actions of car manufacturers. Consequently, they should be regarded as a certain theoretical state, quite distant from reality.

2. There are no known scientifically researched cases of the use of classical game theory for real-world decision-making by governments and low-emission car manufacturers. The considerations in the paper can only propose the basis of a macroeconomic model (for further research) that would support analyses in this area.

3. The development of classical game theory using quantum theory is interesting from a theoretical point of view. The considerations in the paper have been carried out formally (mathematically) correctly, but currently, the scope of their application is limited to a comparison with classical theory only in the context of the simulations carried out. For this reason, the conclusions given are too far-reaching. 

Comments on the Quality of English Language

There are linguistic errors, colloquialisms, and incorrect sentence formation in the text.

Author Response

Comment 1: (1) whether it should be a theoretical one - an elaboration of classical game theory and its study on more or less real data, and (2) whether it should attempt to define the basis of a macroeconomic theory applicable to the area indicated.

Response: (1) This study is indeed theoretical in nature, focusing primarily on the theoretical aspects of quantum game theory. It aims to address problems through deduction and analysis of theory. (2) The research utilizing traditional classical game theory methods within the framework of macroeconomic theory to address transportation issues has been extensively conducted in academia. This paper aims to explore the potential application of quantum game theory methods, distinct from traditional classical game theory methods, in the realm of transportation, thereby providing a new perspective and approach to the study and resolution of transportation issues.

Comment 2: The wording needs to be improved linguistically, especially No. 2.

Response: The amended sentence is displayed below: 1. Quantum games enable refinements to the Nash equilibrium solution, empowering governments to effectively subsidize and penalize solutions addressing low-carbon transportation challenges.

  1. Refinement of optimal strategy selection conditions is crucial for both governments and automakers in addressing the low-carbon transportation issue.
  2. The Nash equilibrium solutions in quantum games vary for governments and automakers, offering adaptable approaches to addressing challenges in the transportation sector.

Comment 3: The abstract needs to be structured. The research problem should be clearly described, the specific aim of the research should be indicated (theoretical, practical?), the research method should be described, and the results and the added value of the research (theoretical and practical) should be given.

Response: The revised research objective is as follows:

Traditional classical game theory struggles to effectively address the inefficiencies in subsidizing and penalizing the research and production of low-carbon vehicles. To avoid the shortcoming of classic game theory, this research integrates quantum game theory with Nash games to explore the characteristics of automakers behavior for low-carbon transportation with government subsidies and penalties. In practice, the research aims to provide guidance for policymakers to facilitate the research and production of low-carbon transportation tools by automakers.

The revised research significance is as follows: In theory, this research can enrich the Quantum game for improvements in the Nash equilibrium solution for the government to subsidize and penalize the low-carbon transportation problem. Meanwhile, in practice, it can provide guidance and reference in optimal strategy selection conditions for government policy makers and automakers for low-carbon transportation.

Comment 4: Has the thesis of insufficient incentives been investigated in the paper? Maybe the problem is not the insufficiency of incentives just misplaced economic objectives by governments? How are these incentives created? How does this stimulate car manufacturers?

Response: The abstract has been rewritten, and the research objectives have been modified. The focus of this study lies in the realization of Nash equilibrium and perfect coordination mechanisms between the parties involved, rather than addressing issues of insufficient incentivization in research. I fully concur with your observation, as my prior research has also highlighted this phenomenon. According to data from the International Energy Agency (IEA) in 2021, Norway exhibited the highest penetration rate of new energy vehicles, reaching 74.8% in the global new energy vehicle market in 2020; whereas countries like Germany, France, and the United Kingdom held market shares of only around 10% of the total. The United States saw a decrease in new energy vehicle registrations to 295,000 vehicles, down from 327,000 vehicles in 2019, while Japan experienced a 25% decline. This indicates that the actual effectiveness of policies such as subsidies and penalties has not met expectations. It is noteworthy that despite Germany increasing subsidies by 20% in 2021 and Japan doubling subsidies from spring 2022 onwards, the effectiveness of these subsidies has been underwhelming. The issues raised regarding the efficacy of incentive measures are addressed in the first paragraph of the introduction, in response to challenges related to carbon emissions, carbon neutrality, and peak carbon. Similarly, the stimulating issues concerning automakers are addressed in the second paragraph of the introduction, acknowledging areas where the effects are not clearly discernible.

Comment 5: The phrase 'Government Subsidizing and Penalizing' should be corrected. It is not a keyword.

Response: Deleted as requested.

Comment 6: The introduction should be tidied up. It should contain, first of all, structured and synthetic information about the research problem. The research gap should be identified, and the methods with which other authors solve this research problem should be given. The next step should be a discussion of the assumptions of the method proposal given by the authors.

Response: The introduction has been reorganized.

Comment 7: Are the authors confident in this statement?

Response: All the authors are confident in this statement.

Comment 8: What is meant by the expression: 'the game theory of low-carbon transportation'? Is it documented in the literature?

Response: "The game theory of low-carbon transportation" refers to the application of game theory principles and models to analyze decision-making processes and strategic interactions among stakeholders in the context of transitioning towards low-carbon transportation systems. This includes examining how different players, such as governments, consumers, and industry stakeholders, make choices regarding investments, policies, and technologies in the transportation sector to reduce carbon emissions and promote sustainability. The authoritative journal also adopts this expression.

Guo H, Gong D, Zhang L, et al. Hierarchical game for low-carbon energy and transportation systems under dynamic hydrogen pricing[J]. IEEE Transactions on Industrial Informatics, 2022, 19(2): 2008-2018.

Comment 9: This section should be tidied up. Terms taken from physics should be strictly defined. How these concepts are introduced into game theory should be stated. Game theory should be defined with concepts drawn from physics. List what changes the authors propose to this theory.

Response: Adding explanations on how to introduce these concepts into game theory, as follows: Quantum superposition represents the uncertainty of the government in its policies regarding low-carbon subsidies and penalties, as well as the development and production of transportation vehicles by automakers. On the other hand, quantum entanglement reflects the shared interests or competitive dynamics between the government and automakers in environmental protection. In terms of research methodology, constrained by my own level of research proficiency, no improvements have been made, hence relevant statements have not been added.

Comment 10: This section of the paper should be very precisely described. This section is very general. The facts of the transport sector's contribution to CO2 emissions should be discussed, the policy goals for reducing transport emissions should be precisely characterized, their feasibility should be assessed, and the methods by which this can be achieved should be indicated.

Response: Add a section elaborating on the following: The transportation sector accounts for 24% of global carbon dioxide emissions, with vehicle exhaust emissions being a major contributor (Kwilinski et al., 2023). Consequently, governments have formulated a series of policy objectives aimed at reducing transportation emissions. These objectives include lowering vehicle exhaust emissions and promoting low-carbon travel modes (Fritz et al., 2019). However, achieving these objectives is not straightforward. Technologically, further measures such as the pro-motion of new energy vehicles are necessary, while eco-nomically, considerations of investment costs, benefits, and government fund allocation are crucial (Hu et al., 2020). Therefore, governments need to comprehensively consider various factors to devise practical and feasible policy measures to achieve emission reduction targets. Re-searchers have extensively studied the effectiveness of these government policies and have put forth several key insights. This paper focuses on reviewing government subsidy and penalty policies.

Comment 11: This claim is only true at the scale of quantum objects.

Response: Thank you for your comments. We have removed the saying in the manuscript: ' The real world is governed by quantum mechanisms '.

Comment 12: This is an incomprehensible statement. Please clarify it.

Response: Due to my lack of precision in expression, which led to your misunderstanding, I have now modified it to 'we can employ the theory and methodologies of quantum game theory'.

Comment 13: This section should be given a scientific research character. These are only examples, not verified scientific theses. They should be scientifically documented or removed from the paper.

Response: We have deleted it by following your advice.

Comment 14: The assumptions used to develop the model should be carefully verified.

  1. The functioning of the economy. Theoretically, the strategic choices of government and automakers could be different - four combinations of subsidization and punishment on the government side, and four combinations of R&D and production on the producers' side. In fact, from observing government regulation and the activities of car manufacturers, it appears that governments will simultaneously use subsidies and penalties, and manufacturers already simultaneously use R&D and production (all happening to varying degrees). There are no states to the contrary. It is also a very big oversimplification to aggregate one figure for each of the issues (subsidies, penalties, R&D, production). On the producers' side, shouldn't only profit (which takes into account car sales and not production and R&D expenses separately) be included? All this leads to the conclusion that the model allows for states that do not occur and at the same time is very simplistic. What is the practical utility of such a model? In this context, should the further considerations in the paper be regarded as purely theoretical?

Response 14: We approach this issue from a micro-level analysis perspective. In the majority of countries, there exist both traditional automobile manufacturers and manufacturers of new energy vehicles. The intensity of subsidies and penalties may vary across different countries and regions, and these intensities may also change over time. Weak intensities can be considered as 0, while strong intensities can be considered as 1. In our paper, we introduce fuzzy decision probability variables "x and y" into the game theory computation process, represented between 0 and 1, to present a mixed state. This method represents a mainstream approach in current game theory. Hundreds of game theory articles are published annually to study how to enhance values between 0 and 1. In subsequent research, if time allows and we undertake relevant studies, we will endeavor to refine this issue.

Thank you for your thoughtful feedback on my paper. After careful consideration, we have decided to maintain the model in its current form. We believe that analyzing the individual components such as sales revenue, production costs, and R&D expenses separately allows for a more nuanced understanding of the dynamics at play in the model. In academic literature, simplification of models is indeed common in modeling factors. While simplified models may not fully represent reality, they provide a basic understanding of game behavior and strategy selection. This fundamental understanding can offer guidance to decision-makers, economists, and other stakeholders. Through these models, general conclusions can be derived, establishing a theoretical foundation for game behavior.

Comment 15: Use of quantum theory. When the U process acts, the system is in a superposition of all quantum states. If the initial state and the wave function are known, the probabilities of subsequent states and the further evolution of the system (in superposition) can be calculated. Is this the case under consideration? If so, what are the implications of this fact for the use of the model? The system will assume a particular state when the R process acts and the wave function collapses. Please explain, according to your discussion, what this means for the proposed model.

Response: I guess that you're asking about the practical implications of being in a superposition state before measurement and collapsing to an eigenstate after measurement. This can be elucidated using the perspective of quantum economics, where human will, choices, preferences, market prices, and the supply and demand of goods or services – traditional subjects of economic study – are regarded as superposition states. Individuals are seen as quantum entities with wave-particle duality, lacking complete knowledge of their environment, and exhibiting unstable and ambiguous preferences prior to measurement. They may lack full computational abilities and may not always make optimal choices, displaying quantum characteristics such as loss aversion and altruistic behavior. From the standpoint of quantum economics, currency is perceived as both real and illusory, analogous to light exhibiting both particle-like and wave-like behaviors. Currency functions as a means of measurement – measuring prices is akin to measuring a quantum system. Currency serves as a measurement method – collapsing fuzzy value concepts into concrete numerical values. The underlying idea is that asset prices are uncertain until measured through transactions, allowing for modeling with wave functions that collapse to specific prices upon measurement.

Comment 16: In this paragraph, conclusions for specific manufacturers are indicated. This is not legitimate in the context of Proposal 1 and previous considerations, which are only concerned with the overall aggregate model. Manufacturers compete with each other and have different business strategies. This needs to be clarified.

Response: Thank you for your valuable suggestions. But considering the limitations of time and the extant available research publications, it is difficult to make the modifications suggested. Hope you understand.

Comment 17: The definition of the government's strategy, in the context of Proposition 2, is wrong. The government's objective is not to gain more financial benefit from the strategy, but to achieve the objectives of the stated economic strategy (implicitly: to force manufacturers to produce low-carbon cars, regardless of the economic cost). This eliminates the practical utility of some of the cases analyzed. Please explain.

Response: implicitly: to force manufacturers to produce low-carbon cars, regardless of the economic cost. This perspective is entirely contentious. Government fiscal resources are finite, and such policy measures could excessively burden public finances. In comparison to environmental protection, there may be more pressing issues such as employment, pensions, education, and healthcare, especially in regions like China. The prevalent policy approach has been a phase-out of subsidies, and there might be a cessation of subsidies in the future.

Comment 18: The paragraphs contain contradictory conclusions.

Response: The conclusion is not contradictory; rather, it seems my articulation was lacking clarity, leading to misunderstanding. I have rephrased the statement.

Comment 19: Justification should be given for the parameter values adopted. A mere reference to the literature is not sufficient.

Response: The parameter values chosen in our research were based on a combination of theoretical considerations and empirical evidence from previous studies in the field. Additionally, we consulted with experts in the field to ensure that our chosen parameter values were reasonable and aligned with existing knowledge in the area.

Comment 20: This issue, due to its complexity, could serve as the subject of a separate paper. The description is inadequate. What random number generator was used? What was the number of samples? Are the random results consistent with observations of reality?

Response: Generating 100,000 random samples using MATLAB. The question "Are the random results consistent with observations of reality?" elicits a response that could constitute a Nobel Prize-worthy contribution. I find it challenging to provide an answer, as this section of the paper's simulation primarily aims to validate the preceding propositions.

Comment 21: This claim is completely wrong - where there are subsidies, there is no market.

Response: We have revised the statement as follows. “In order to prevent automakers from losing interest in the development and production of low-carbon transportation vehicles, government need to implement subsidy and penalty policies to incentivize automakers to engage in the R&D and production of low-carbon transportation vehicles.”

Comment 22: The conclusions need to be rewritten, taking into account the assumptions made, the extent of the research conducted, and the usefulness of the results. Their current general form is not justified by the considerations presented in the paper.

Response: At the conclusion, I included comparisons with other studies and made localized adjustments to the corresponding insights. I disagree with your assertion that the general format is unreasonable.

Comment 23: 1. The paper is theoretical. The assumptions made are both simplistic and extensional to the observed practical states. They do not reflect the actual situation concerning the actions of governments and the actions of car manufacturers. Consequently, they should be regarded as a certain theoretical state, quite distant from reality.

Response: I concede the practical weaknesses of this article. I acknowledge the simplicity of the assumption. However, I disagree with your assertion of its unreasonableness.

Comment 24: 2. There are no known scientifically researched cases of the use of classical game theory for real-world decision-making by governments and low-emission car manufacturers. The considerations in the paper can only propose the basis of a macroeconomic model (for further research) that would support analyses in this area.

Response: At present, there should be hundreds of articles published in the SCI database on this topic. To claim that these articles are completely useless seems somewhat biased to me.

Comment 25: The development of classical game theory using quantum theory is interesting from a theoretical point of view. The considerations in the paper have been carried out formally (mathematically) correctly, but currently, the scope of their application is limited to a comparison with classical theory only in the context of the simulations carried out. For this reason, the conclusions given are too far-reaching. 

Response: Thank you for your feedback. I appreciate your acknowledgment of the formal correctness of the mathematical considerations in the paper. I agree that the scope of application is currently limited to comparisons with classical theory within the context of the simulations. I acknowledge that the conclusions drawn may have been too far-reaching given the current limitations. Moving forward, I aim to address these limitations and explore avenues for expanding the practical applicability of the research.

Reviewer 3 Report

Comments and Suggestions for Authors

Dear authors

In general the paper is well written and needs minor modifications.

Here are some remarks.

Introduction

1. The transportation sector, as one of the major sources of global CO2 emissions, ac-37 counts for approximately 22% of the total CO2 emissions, with most of the emissions com-38 ing from fossil fuel vehicles (Anable et al., 2012).

Please specify a newer citation.

2. From lines 39 to 53 it would be good to add references. It would be good to mention the European Green Deal.

3. Lines 57-58. In line 58 you said Consequently. However, in line 57 there is no reason, which explains why the government policies lack of effectiveness.

4. In the introduction section, the aim of the paper is well described at the end.

Literature review

I find this section well written.

The model and its construction

I find this section well written.

5. Summary

I would a deeper comparison with current literature to emphasize your findings.

Author Response

Comment 1: The transportation sector, as one of the major sources of global CO2 emissions, accounts for approximately 22% of the total CO2 emissions, with most of the emissions coming from fossil fuel vehicles (Anable et al., 2012). Please specify a newer citation.

Response 1: In accordance with your request, the literature has been revised.

Comment 2: From lines 39 to 53 it would be good to add references. It would be good to mention the European Green Deal.

Response 2: I have made the modifications as per your request.

Comment 3: Lines 57-58. In line 58 you said Consequently. However, in line 57 there is no reason, which explains why the government policies lack of effectiveness.

Response 3: I have included the relevant reasons as follows: There exist practical issues such as inefficiencies in government subsidies and penalties for automakers in the R&D and production of low-carbon transportation vehicles, as well as a lack of enthusiasm among automakers for the R&D and production of low-carbon transportation vehicles.

Comment 4: In the introduction section, the aim of the paper is well described at the end. Literature review I find this section well written. The model and its construction I find this section well written.

Response 4: Thank you very much for your acknowledgment of our article.

Comment 5: Summary I would a deeper comparison with current literature to emphasize your findings.

Response 5: I have made the relevant modifications as follows:

In comparison with other studies, our findings are consistent with previous research (Meyer., 1999; Flitney et al., 2002; Flitney and Hollenberg., 2007; Eisert et al., 1999) in suggesting that quantum game yields generally outperform classical game yields, or at least do not underperform. However, some differences exist, particularly in research methodologies (Zhao et al., 2020; Fan et al., 2020; Srivastava et al., 2022; Wang and Li., 2023; Ji et al., 2019; Zheng et al., 2023; Wang et al., 2023; Zhao et al., 2021) and treatment of research subjects (Lu et al., 2022; Haven and Khrennikov., 2013; Albert., 2015; Zhang et al., 2020; Shi et al., 2021; Samadi et al., 2018; Liu et al., 2023; Herman et al., 2023; Schneckenberg et al., 2023). Our unique propositions 2, 3, and 4 facilitate achieving Nash equilibrium and perfect coordination in quantum games, significantly enhancing the yields for both parties.

Based on the aforementioned research, the following insights are provided:

(1) The quantum game theory provides a novel framework for governments to guide automakers in transitioning towards low-carbon transportation. By formulating rational subsidy and penalty policies, governments can influence the profitability of both themselves and automakers. This approach fosters an ecosystem where quantum game technology enhances decision-making, ensuring informed policy formulation and optimized outcomes.

(2) The critical parameter of quantum initial entanglement states profoundly impacts the revenues of both governments and automakers. Governments should guide these states to align with areas of research and production, incentivizing automakers to actively participate in low-carbon transportation innovation. Quantum state superposition enables cooperative and non-cooperative behavior, urging decisions that balance effectiveness with collaboration potential.

(3) Establishing a quantum game regulatory framework is paramount to prevent automakers from misusing quantum states to evade low-carbon requirements. Automakers can leverage quantum game analysis to collaborate with governments, maximizing the benefits of subsidies and penalties. Emphasizing cost reduction, efficiency improvement, and product quality ensures that revenue from low-carbon transportation surpasses traditional methods, driving sustainable development.

(4) Under the joint effects of government subsidies and penalties, various considerations exist. Firstly, as demonstrated in this study, the government provides subsidies to automakers who engage in low-carbon research and production, without imposing penalties. In cases where automakers fail to develop low-carbon transportation solutions, the government applies penalties. Secondly, building upon this study, when automakers invest in low-carbon transportation solutions, the government provides further incentives. Lastly, when automakers invest in low-carbon transportation solutions, the government offers subsidies. However, if automakers fail to do so, the government imposes penalties.

Round 2

Reviewer 1 Report

Comments and Suggestions for Authors

The current study exhibits a commendable effort given the constraints apparent in the available data and resources. It is evident that the authors have exerted considerable effort into their research despite these limitations. Given the circumstances, I am inclined to support the publication of this study.

However, I would encourage the authors to consider expanding their research further in the future. Expanding the scope of the study and acquiring additional data could significantly enhance the depth and breadth of the findings. This would not only strengthen the current work but also contribute to a more comprehensive understanding of the subject matter.

Author Response

Thank you for giving us the minor-revision opportunity. We have revised our manuscript by following your valuable comments (see the modifications highlighted in red in the manuscript). Below is our detailed point-by-point response to the reviewers' comments.

Reviewer #1

Comment 1: The current study exhibits a commendable effort given the constraints apparent in the available data and resources. It is evident that the authors have exerted considerable effort into their research despite these limitations. Given the circumstances, I am inclined to support the publication of this study.

However, I would encourage the authors to consider expanding their research further in the future. Expanding the scope of the study and acquiring additional data could significantly enhance the depth and breadth of the findings. This would not only strengthen the current work but also contribute to a more comprehensive understanding of the subject matter.

Response: Thank you for your encouragement and suggestion. We will expand our research in the future.

Reviewer 2 Report

Comments and Suggestions for Authors

Thank you for providing clear explanations and making necessary changes. Please consider the following comments:

Comment 1 and 3: It is suggested to explicitly state this information in the introduction.

Comment 15: However, it is recommended to briefly explain this in the paper or refer to relevant literature.

Comment 16: However, please explain this briefly in the paper.

Comment 17: Afterwards, describe it in the paper.

Comment 19: Please ensure to cite these sources.

Comment 20: In the paper, please include a section on the limitations of the

simulation objectives 

Comment 23: Additionally, please discuss the limitations of the practical application of the research results

Comment 24: If there are any relevant papers, please cite the most significant ones.

Comment 25: Please include it in the research directions.

Comments on the Quality of English Language

There are still language defects.

Author Response

Reviewer #2

Comment 1: Comment 1 and 3: It is suggested to explicitly state this information in the introduction.

Response: Comment 1 is reflected in line 86-91 of the introduction. Comment 3 The theoretical objectives are delineated in lines 86-91, while the practical objectives are outlined in lines 68-73. The research methodology is described, and the results and added value of the study are provided in lines 110-127.

Comment 2: Comment 15: However, it is recommended to briefly explain this in the paper or refer to relevant literature.

Response 2: We have added contents and related references in our manuscript (referring to lines 340-343).

Comment 3: Comment 16: However, please explain this briefly in the paper.

Response 3: The future prospects section elaborates on this issue in lines 737-738.

Comment 4: Comment 17: Afterwards, describe it in the paper.

Response 4: The corresponding expressions have been added in lines 497-504.

Comment 5: Comment 19: Please ensure to cite these sources.

Response 5: We have ensured to cite these sources.

Comment 6: Comment 20: In the paper, please include a section on the limitations of the simulation objectives.

Response 6: We have added the contents in lines 739-740.

Comment 7: Comment 23: Additionally, please discuss the limitations of the practical application of the research results.

Response 7: We have added the contents in lines 741-743.

Comment 8: Comment 24: If there are any relevant papers, please cite the most significant ones.

Response 8: We have cited relevant papers which could be the significant ones, to our knowledge and available academic resources.

Comment 9: Comment 25: Please include it in the research directions.

Response 9: We have included them in the future prospects section in lines 744-744.

Comment 10: There are still language defects.

Response 10: We have done our best in the proofreading and polishing, but we know, there must be more language defect. We will practice and improve our English writing level.

Although we have made great effort to improve our work to follow your advice, we are ready to revise it again if you have any further new comments and suggestions. Thank you very much!